# Connexin43 promotes exocytosis of damaged lysosomes through actin remodelling

Neuza Domingues[1,2,3,4,5], Steve Catarino [1,2,3,4], Beatriz Cristóvão[1,2,3,4], Lisa Rodrigues [6],
Filomena A Carvalho [7], Maria João Sarmento [7], Mónica Zuzarte[1,2,3,4], Jani Almeida [2,3,4,6],
Teresa Ribeiro-Rodrigues [1,2,3,4], Ânia Correia-Rodrigues [1,2,3,4], Fábio Fernandes[8],
Paulo Rodrigues-Santos [2,3,4,6], Trond Aasen [9], Nuno C Santos[7], Viktor I Korolchuk[10],
Teresa Gonçalves [2,3,6], Ira Milosevic[5,11], Nuno Raimundo [5,12] & Henrique Girão [1,2,3,4]✉

## Abstract

A robust and efficient cellular response to lysosomal membrane damage prevents leakage from the lysosome lumen into the cytoplasm. This response is understood to happen through either lysosomal membrane repair or lysophagy. Here we report exocytosis as a third response mechanism to lysosomal damage, which is further potentiated when membrane repair or lysosomal degradation mechanisms are impaired. We show that Connexin43 (Cx43), a protein canonically associated with gap junctions, is recruited from the plasma membrane to damaged lysosomes, promoting their secretion and accelerating cell recovery. The effects of Cx43 on lysosome exocytosis are mediated by a reorganization of the actin cytoskeleton that increases plasma membrane fluidity and decreases cell stiffness. Furthermore, we demonstrate that Cx43 interacts with the actin nucleator Arp2, the activity of which was shown to be necessary for Cx43-mediated actin rearrangement and lysosomal exocytosis following damage. These results define a novel mechanism of lysosomal quality control whereby Cx43-mediated actin remodelling potentiates the secretion of damaged lysosomes.

**Keywords** Lysosomal Damage; Exocytosis; Connexin43; Arp2; Actin-remodelling
**Subject Categories** Cell Adhesion, Polarity & Cytoskeleton; Membranes & Trafficking; Organelles

## Introduction

Lysosomes are essential degradative organelles that confer cells with the ability to act autonomously by producing their own energy supply as well as providing a regenerative and defence capacity. More than a mere degradation structure, these acidic organelles have been associated with a plethora of biological processes, such as signalling hubs, pathogen killing, exocytosis, plasma membrane repair, lipid homeostasis and cell death (Blott and Griffiths, 2002; Carroll and Dunlop, 2017; de Duve, 2005; Luzio et al, 2007; Wartosch et al, 2015). Lysosomes are exposed to a myriad of stressors that can affect the integrity of their membrane, including intracellular reactive oxygen species (Pascua-Maestro et al, 2017; Pivtoraiko et al, 2009), aggregates (Umeda et al, 2011), and extracellular agents, such as pathogens (Niekamp et al, 2022; Radulovic and Stenmark, 2020; Westman et al, 2018), lysosomo-tropic drugs (Radulovic and Stenmark, 2020), or silica crystals (Maejima et al, 2013). Indeed, loss of lysosomal membrane integrity does not only affect cellular clearance capacity but also induces the release into the cytosol of multiple factors, such as cathepsins and calcium ($Ca^{2+}$), with subsequent activation of several forms of cell death (Boya and Kroemer, 2008; Bröker et al, 2004; Radulovic and Stenmark, 2020). Thus, a fine-tuned control of the physical integrity and function of lysosomes is fundamental to maintain cell homeostasis, function and fate.

Several studies have elucidated the complexity of cellular and molecular mechanisms implicated in the response to lysosomal damage. $Ca^{2+}$ release from small perforations of the lysosomal membrane is sensed by ALG2-interacting protein X (Alix) and its $Ca^{2+}$-binding partner apoptosis-linked gene-2 (ALG2). This triggers the rapid recruitment of the endosomal sorting complex required for transport (ESCRT)-III (Radulovic et al, 2018; Skowyra et al, 2018), to seal the damaged membrane. Alternatively, when repair mechanisms are not sufficient to counteract the damage,

[1]Univ Coimbra, Coimbra Institute for Clinical and Biomedical Research (iCBR), Faculty of Medicine, Coimbra, Portugal. [2]Univ Coimbra, Faculty of Medicine, Coimbra, Portugal. [3]Univ Coimbra, Centre for Innovative Biomedicine and Biotechnology (CIBB), Coimbra, Portugal. [4]Clinical and Academic Centre of Coimbra, Coimbra, Portugal. [5]Multidisciplinary Institute of Ageing, University of Coimbra, Coimbra, Portugal. [6]Univ Coimbra, Center for Neurosciences and Cell Biology (CNC), Coimbra, Portugal. [7]Instituto de Medicina Molecular, Faculdade de Medicina, Universidade de Lisboa, Lisboa, Portugal. [8]Institute for Bioengineering and Biosciences (IBB) and Associate Laboratory i4HB-Institute for Health and Bioeconomy, Department of Bioengineering, Instituto Superior Técnico, Universidade de Lisboa, Lisbon, Portugal. [9]Vall d'Hebron Research Institute (VHIR), Barcelona, Spain. [10]Biosciences Institute, Faculty of Medical Sciences, Newcastle University, Newcastle, UK. [11]University of Oxford, Centre for Human Genetics, Nuffield Department of Medicine, Oxford, UK. [12]Department of Cellular and Molecular Physiology, Penn State College of Medicine, Hershey, PA, USA. ✉E-mail: hmgirao@fmed.uc.pt

lysosomes are tagged for degradation through lysophagy. This specialized form of autophagy is initiated by the recruitment of cytosolic galectins such as galectin-3 (Gal3) and Gal9, which bind to luminal glycans from the exposed glycosylated lysosomal proteins (Aits et al, 2015). These sensor molecules trigger the ubiquitination of lysosomal proteins (Fujita et al, 2013; Koerver et al, 2019), with the formation of autophagosomes and their fusion with intact lysosomes for degradation (Maejima et al, 2013; Papadopoulos, et al, 2020). Injured lysosomes were also shown to potentiate cellular secretion (Burbidge et al, 2022), suggesting an intricate interplay between repair, lysophagy and exocytosis systems when lysosomal membrane integrity is lost (Sivaramak-rishnan et al, 2012). However, the mechanisms and players that mediate this crosstalk remain largely elusive.

Cx43 is a monomeric transmembrane protein that oligomerizes to form hexameric channels allowing the exchange of information between adjacent cells. Besides this canonical function, Cx43 contributes to extracellular vesicle-mediated long range communication (Soares et al, 2015; Martins-Marques et al, 2020, 2019; Ribeiro-Rodrigues et al, 2017), multivesicular body (MVB) loading (Martins-Marques et al, 2022), mitochondria (Fu et al, 2017; Shimura and Shaw, 2022; Shimura et al, 2021) and nucleus maintenance (Dang et al, 2003; Epifantseva et al, 2020; Martins-Marques et al, 2023). In addition, this ubiquitous Cx has been implicated in regulating protein (Agullo-Pascual et al, 2014; Basheer et al, 2017) and organelle localization (Fu et al, 2017), as well as membrane fission events through the modulation of actin structures (Shimura et al, 2021; Shimura and Shaw, 2022). Therefore, it is conceivable that the presence of Cx43 in the endocytic pathway goes beyond a mere lysosomal substrate and is instead an active player of this trafficking process. In the present study, we aimed to address the non-canonical role of Cx43 in lysosomal homeostasis. Here we assigned to Cx43 a function during the cellular response to lysosomal damage. We show that Cx43 is redistributed to the membrane of damaged lysosomes and assists in actin network remodelling to allow their access to the plasma membrane. We postulate that Cx43 is involved in membrane quality control mechanisms, contributing to cell recovery from lysosomal membrane permeabilization.

# Results

## Exocytosis is involved in cellular response to lysosomal injury

To address the role of exocytosis as part of the cellular response to lysosomal damage, we used a well-established protocol to induce lysosomal damage through the lysosomotropic agent L-leucyl-L-leucine methyl ester (LLOMe). To confirm the effectiveness of this model, we first assessed the effect of LLOMe treatment in HEK293A cells transfected with mCherry-Gal3. In agreement with previous studies (Jia et al, 2020b; Skowyra et al, 2018), after 10 min of incubation with LLOMe, the formation of mCherry-Gal3-positive small puncta starts to be visible (Fig. S1a), increasing in intensity after 60 min (Appendix Fig. S1b). Notably, a redistribution of LAMP1 and Gal3 positive vesicles towards the cell periphery was also observed (Appendix Fig. S1b). Since the peripheric lysosomal distribution has been associated with increased

exocytosis (Jaiswal et al, 2002), we next investigated the link between lysosomal membrane permeability and lysosome secretion (Fig. 1A). We started by determining the presence of LAMP1 at the plasma membrane, through flow cytometry, which is a widely used approach to assess lysosomal fusion with the cell surface. Our data show a gradual increase in LAMP1 intensity at the cell surface after 10 and 60 min of LLOMe treatment, which was maintained at 180 min (Fig. 1B,C). Cells loaded with $Ca^{2+}$ and the $Ca^{2+}$-ionophore ionomycin were used as a positive control of lysosomal exocytosis (Rodríguez et al, 1997). To complement these results, we also measured the release of lysosomal content, namely the enzyme cathepsin B (Cath B), into the extracellular milieu. Accordingly, cells treated with either ionomycin-$Ca^{2+}$ or LLOMe, secreted significantly higher levels of Cath B (Fig. 1D). Furthermore, we performed cell-surface protein biotinylation assays to investigate whether LLOMe treatment increased the presence of damaged lysosome markers, such as Gal3, at the cell surface. Results presented in Fig. 1E and Appendix Fig. S1c,d show that the levels of biotinylated Gal3 and LAMP1 increased in LLOMe-treated cells.

It is widely accepted that cells respond to lysosome rupture by activating repair mechanisms and/or by targeting lysosomes for degradation through lysophagy (Fig. 1A). To explore the possibility of lysosomal exocytosis as an alternative disposal route for damaged lysosomes, lysosome repair was impaired through the knockdown of HEK293A Alix (Alix[KD]) (Appendix Fig. S1e), while lysophagy was impaired through the knockdown of Atg7 (Atg7[KD]) (Appendix Fig. S1f) or the knockout of all six human Atg8 proteins (Atg8[KO]) (Appendix Fig. S1g). Results depicted in Fig. 1F–I show that in cells treated with LLOMe, silencing of either Alix or Atg7 led to an increase in cell surface levels of LAMP1, with augmented amounts of secreted Cath B (Appendix Fig. S1h,i). Accordingly, HeLa Atg8[KO] cells also presented increased levels of lysosomal exocytosis after LLOMe treatment (Appendix Fig. S1j,k). Of note, Gal3 puncta were visible in Alix[KD] and Atg7[KD] cells following 60 min of LLOMe treatment, indicating that impairment of lysosomal repair or degradation is not affecting the recruitment of Gal3 to injured lysosomes and results in the accumulation of injured lysosomes (Appendix Fig. S1l,m), as previously reported (Maejima et al, 2013; Skowyra et al, 2018), as well as in Atg8[KO] HeLa cells (Appendix Fig. S1n). These observations were further corroborated through pharmacological approaches. To impair lysosomal repair by ESCRT machinery, we used the $Ca^{2+}$ chelator BAPTA (Jia et al, 2020b; Skowyra et al, 2018), whereas to block autophagy, cells were incubated with the phosphoinositide 3-kinase (PI3K) inhibitor, 3-methyladenine (3MA) (Wu et al, 2010). Treatment with any of these drugs alone did not lead to significant changes in LAMP1 levels at the cell surface (Appendix Fig. S1o). However, in the presence of LLOMe, treatment with either BAPTA and/or 3MA further increased cell surface levels of LAMP1 (Appendix Fig. S1p,q) and Cath B secretion (Appendix Fig. S1r). Importantly, the efficiency of both treatments was confirmed by the inhibition of Alix recruitment to damaged lysosomes upon BAPTA treatment and the reduction of LLOMe-induced LC3-II accumulation by 3MA (Appendix Fig. S1s,t). Thus, an increase in lysosomal membrane permeability seems to be tightly linked with exocytosis activation, which is potentiated by the impairment of repair or lysophagy mechanisms.

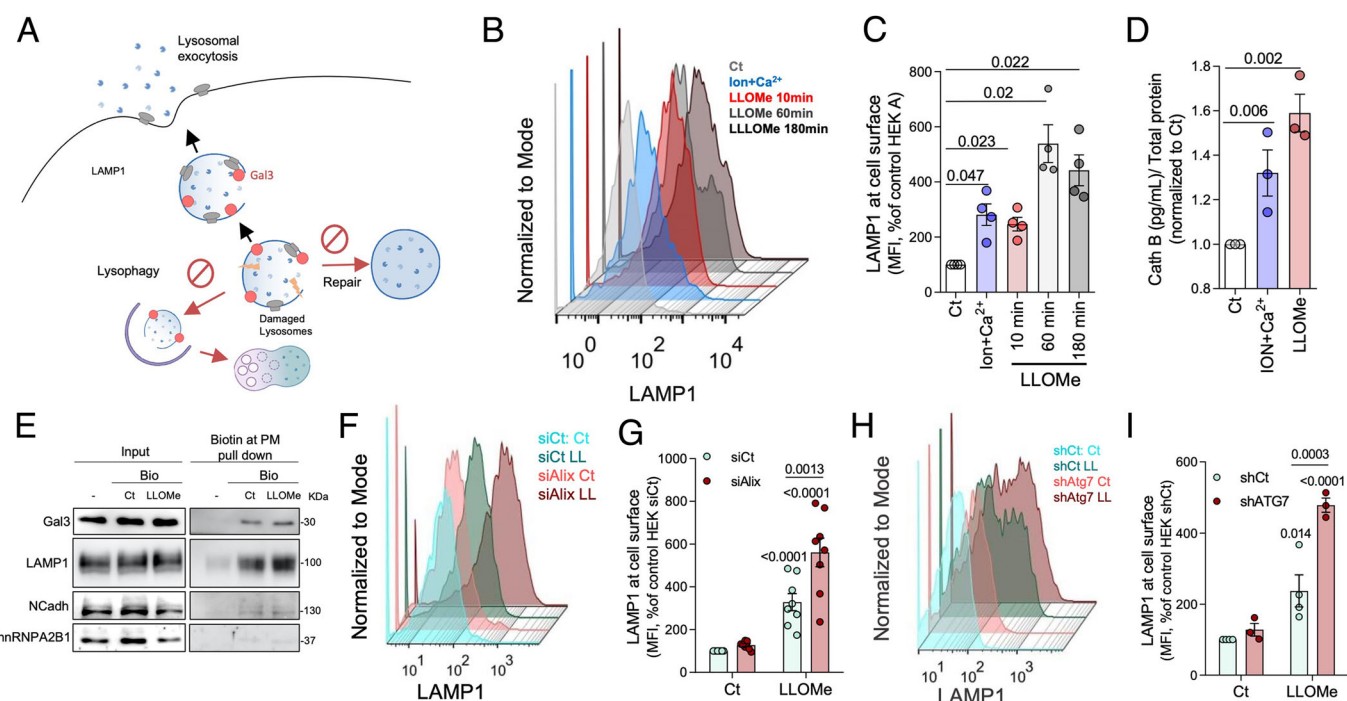

**Figure 1. Lysosome injury triggers exocytosis.**

(A) Schematic outline of our experimental strategy and the intervention points to evaluate exocytosis as an alternative to repair or degradation of damaged lysosomes. (B) Representative histogram of the mean intensity of cell surface LAMP1 measured by flow cytometry in control cells, cells stimulated with ionomycin (Ion) and calcium ($Ca^{2+}$) for 10 min, or several incubation times of LLOMe. (C) Quantification of the mean intensity of the cells from condition (B). Data are means ± SEM ($n = 4$). $P$ values were calculated by one-way ANOVA with Dunnett's multiple comparisons test. (D) Quantification of cathepsin B (Cath B) release in control, ionomycin and $Ca^{2+}$ or LLOMe-treated cells for 1 h. Data are means ± SEM ($n = 3$). $P$ values were calculated by one-way ANOVA with Tukey's multiple comparisons test. (E) Immunoblot analysis of plasma membrane (PM) biotinylated proteins (Bio) from control and cells treated with LLOMe for 60 min. Immunoblot of N-cadherin (NCadh) and heterogeneous nuclear ribonucleoprotein A2/B1 (hnRNPA2B1) were used as controls. (F) Representative histogram of the mean intensity of LAMP1 at cell surface measured by flow cytometry in control or cells silenced to Alix (siAlix) and treated with LLOMe. Controls were performed with transfection of scramble siRNA (siCT). (G) Quantification of the mean intensity of LAMP1 at cell surface from (F). Data are means ± SEM ($n = 8$). (H) Histogram of the mean intensity of cell surface LAMP1 measured by flow cytometry in cells depleted of Atg7 (shAtg7) and treated with LLOMe for 60 min. Control cell line was infected with a virus expressing a scramble shRNA (shCT). (I) Quantification of the mean intensity of LAMP1 at cell surface from (H). Data are means ± SEM ($n = 3$). $p$ values were calculated by two-way ANOVA with Sidak's multiple comparisons test in (C, D, G, I). Source data are available online for this figure.

## Cx43 potentiates lysosomal exocytosis

Lysosome secretion involves a series of highly coordinated sequential steps, which include cytoskeleton reorganization and membrane fusion events. Compelling evidence have implicated Cx43 in actin network reorganization (Basheer et al, 2017; Chen et al, 2015; Fu et al, 2017; Smyth et al, 2012) and interaction with membrane fusion machinery, including ESCRT complex proteins, such as TSG101 and Hrs (Auth et al, 2009; Leithe et al, 2009). Given the regulatory role of Cx43 in diverse biological processes allied to its accumulation in lysosomes (Leithe et al, 2009), we hypothesized that Cx43 can play a role in lysosome secretion following damage. To address this question, we used a HEK293A cell line stably expressing Cx43, HEK293[Cx43+], which display a similar phenotype and behaviour to cells that endogenously express Cx43 (Catarino et al, 2011; Martins-Marques et al, 2020; Ribeiro-Rodrigues et al, 2014). First, we investigated the impact of Cx43 expression on LLOMe-induced lysosomal damage by comparing the formation of mCherry-Gal3 puncta in HEK293[Cx43+] and the HEK293A parental cell line, which present negligible levels of endogenous Cx43 (Appendix Fig. S2a). First, we showed that Cx43

expression does not affect recruitment of Gal3 upon lysosome damage (Appendix Fig. S2b,c). We then conducted a kinetic study to monitor cellular recovery after treatment with LLOMe for 60 min (Fig. 2A; Appendix Fig. S2c). Notably, after the first hour of LLOMe washout, Gal3 puncta levels were higher in HEK293A cells, when compared to cells overexpressing Cx43. These results were confirmed in SUM159PT cells (SUM[WT]), a breast cancer cell line, where Cx43 KO (SUM[Cx43KO]) led to a significant increase in the intensity of Gal3 puncta over the washout time, suggesting a reduced recovery ability to endomembrane injury when compared to parental cells expressing endogenous levels of Cx43 (Appendix Fig. S2d,e). This result reinforces the notion that Cx43 modulates cell response to LLOMe-induced lysosomal damage.

To unveil the mechanism behind this observation we assessed the impact of elevated Cx43 levels in lysosomal homeostasis and autophagy. Our results demonstrated that Cx43 expression had no significant effect on total levels of lysosomal proteins, such as LAMP1 and Cath B (Appendix Fig. S3a–h), on lysosomal pH (Appendix Fig. S3i) and its degradative activity (Appendix Fig. S3j). In addition, we did not observe an effect on autophagic capacity, with no significant differences on LC3 levels and autophagic flux

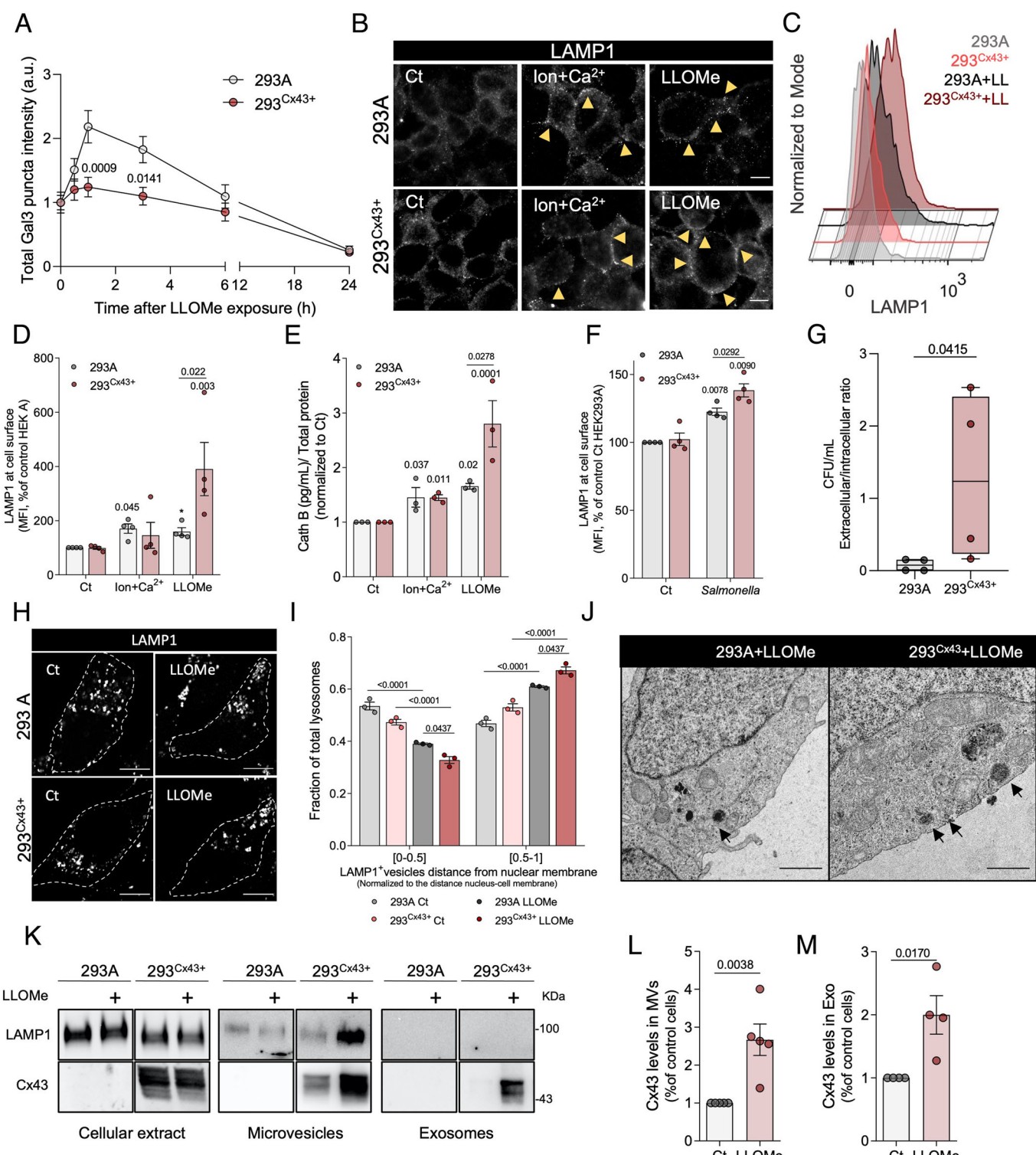

(Appendix Fig. S4a–e), or on other autophagy-related proteins such as p62 and Beclin-1 (Appendix Fig. S4f–j), neither in control conditions nor following LLOMe treatment. This suggests that lysosomal biogenesis and autophagy are not affected by Cx43 levels. Next, we evaluated the effect of Cx43 expression on lysosome exocytosis in response to LLOMe treatment. Remarkably,

HEK293$^{Cx43+}$ cells presented higher LAMP1 levels at the cell surface (Fig. 2B–D) and a significant increase in Cath B secretion (Fig. 2E) after lysosomal damage, when compared to HEK293A. Furthermore, the LLOMe-triggered increase in LAMP1 at the plasma membrane was diminished in Cx43 knocked out cells (Appendix Fig. S2f). We also observed that Cx43 did not potentiate

**Figure 2. Cx43 potentiates exocytosis after lysosomal damage.**

(A) Time-course plot of total mcherry-Galectin3 (Gal3) intensity in HEK293A or HEK293$^{Cx43+}$ cells after LLOMe washout. Cells were treated with LLOMe for 60 min and then incubated with fresh medium for the indicated times. Cells recovery was assessed by quantification of total Gal3 intensity by fluorescence imaging of fixed cells. Values were normalized to the mean intensity of Gal3 after LLOMe treatment. Data are means ± SEM ($n = 3$). At least 15 cells were quantified per experimental condition for each independent experiment. (B) Representative immunofluorescence images of cell surface LAMP1 in control HEK293A or HEK293$^{Cx43+}$ cells, cells treated for 10 min with ionomycin and calcium or with LLOMe during 60 min. The LAMP1 staining was performed in non-permeabilized live cells on ice. Yellow arrows point to LAMP1 accumulation at the plasma membrane. Scale bar, 10 µm. (C) Histogram representation of the mean intensity of LAMP1 at cell surface measured by flow cytometry in HEK293A or HEK293$^{Cx43+}$ control cells or treated for 60 min with LLOMe (LL). Data are means ± SEM ($n = 3$). (D) Quantification of the mean intensity of LAMP1 at cell surface from HEK293A or HEK293$^{Cx43+}$ cells from the same conditions as (B). Data are means ± SEM ($n = 4$). (E) Cathepsin B (Cath B) release by HEK293A or HEK293$^{Cx43+}$ cells from the same conditions as (B). Data are means ± SEM ($n = 3$). (F) Quantification of the mean intensity of cell surface LAMP1 from HEK293A or HEK293$^{Cx43+}$ cells infected with *Salmonella* for 60 min. Data are means ± SEM ($n = 4$). (G) Boxplot of the ratio values between the number of colony forming units (CFU) in the supernatant and in the cellular fraction (corresponding to intracellular bacteria) at 3 h post-infection. Data are means ± SEM ($n = 4$). (H) Immunofluorescence images of LAMP1 in HEK293A or HEK293$^{Cx43+}$ control cells or after 60 min of LLOMe treatment. Dashed lines indicate cell edges. Scale bar, 10 µm. (I) Fraction quantification of lysosomes as a function of distance from the nuclear membrane. The distance between each lysosome and nuclear membrane was determined in function of the total distance between the nucleus and the plasma membrane. At least 15 cells were analysed per condition per each independent experiment. Data are means ± SEM ($n = 3$). (J) Transmission electron microscopy of HEK293A or HEK293$^{Cx43+}$ cells treated with LLOMe for 60 min. Scale bar, 15 µm. (K) Immunoblot analysis of LAMP1 and Cx43 levels in total extract and extracellular vesicles released into the medium, including microvesicles (MVs) and exosomes (Exo) of HEK293A or HEK293$^{Cx43+}$ cells treated with LLOMe for 60 min. (L, M) Quantification of Cx43 protein levels in microvesicles (MVs, $n = 5$) (L) and exosomes (Exo, $n = 4$) (M) isolated from HEK293$^{Cx43+}$ control cells or after LLOMe treatment. Data are means ± SEM. $p$ values were calculated by Two-way ANOVA with Tukey's multiple comparisons test in (A, D, E, F, I), unpaired two-tailed t test in (L, M) and Kolmogorov–Smirnov test in (G). Source data are available online for this figure.

lysosomal secretion in response to increased intracellular Ca$^{2+}$ following ionomycin-Ca$^{2+}$ treatment (Fig. 2B,D).

Because phagolysosomes containing pathogens induce endomembrane rupture (Domingues et al, 2022; Herbst et al, 2020), we assessed whether Cx43 would potentiate the secretory ability of epithelial cells after infection with *Salmonella enterica* (Niekamp et al, 2022). To address this question, we measured LAMP1 at the cell surface and, in parallel, the number of *Salmonellas* secreted by HEK293A and HEK293$^{Cx43+}$ after infection by measuring the number of colony forming units (CFUs) in the intra- and extracellular fraction. Our data revealed that infected HEK293$^{Cx43+}$ cells have increased LAMP1 at the plasma membrane when compared with HEK293A (Fig. 2F). Accordingly, we also observed an augment in the ratio between the numbers of expelled *Salmonella* present in the supernatant and the levels of intracellular pathogens after 3 h of infection (Fig. 2G). Thus, the presence of Cx43 increased the ability of cells to expel *Salmonella*, presumably due to the secretion of lysosomes damaged by these pathogens.

Furthermore, considering that peripheral lysosomes are involved in the secretion process (Jaiswal et al, 2002), we investigated whether Cx43 expression affected lysosome distribution. The results depicted in Fig. 2H,I show that while LLOMe treatment increased the total number of peripheral lysosomes in both cell lines, this event was exacerbated in Cx43 expressing cells. This result was supported by transmission electron microscopy (TEM) imaging, where we observed more lysosomes near the plasma membrane of HEK293$^{Cx43+}$ after LLOMe treatment when compared with HEK293A (Fig. 2J). In basal conditions, HEK293$^{Cx43+}$ cells presented a normal distribution of lysosomes.

Considering our previous studies showing that EV are enriched in Cx43 (Soares et al, 2015; Martins-Marques et al, 2022), and other findings associating lysosomal dysfunction with increased extracellular vesicle (EV) release (Miranda et al, 2018), we next evaluated the effect of LLOMe-induced lysosomal damage on the levels of Cx43 secreted in EV. Results presented in Fig. 2K–M demonstrate that LLOMe treatment led to a substantial increase in the amount of Cx43 released in both microvesicle and exosome fractions, without a significant effect on vesicle size (Appendix Fig. S2g).

Interestingly, Cx43 expression enhanced the release of LAMP1 in microvesicles following lysosomal damage. Altogether, these results suggest that Cx43 has a role in coordinating the cellular response to either LLOMe- or pathogen-triggered lysosomal damage by potentiating their secretion.

## Cx43 is recruited to damaged lysosomes

Considering the diverse subcellular localization of Cx43 (Martins-Marques et al, 2019; Ribeiro-Rodrigues et al, 2017; Soares et al, 2015), we next investigated the effect of lysosomal damage on Cx43 distribution. Through fluorescence microscopy assays, we showed that LLOMe treatment led to a partition of Cx43 to Gal3 positive lysosomes (Movie EV1, Fig. 3A, Appendix Fig. S5a for control cells), suggesting the recruitment of Cx43 to damaged lysosomal membranes (Fig. 3b; Appendix Fig. S5b). In addition, we observed a marked increase in colocalization of Cx43 and LAMP1 in LLOMe-treated cells (Fig. 3C). Notably, ~25% of LAMP1-positive vesicles contained both Cx43 and Gal3 (Appendix Fig. S5c). Moreover, TEM immunogold imaging confirmed the presence of Cx43 and LAMP1 on the lysosomal membrane of isolated lysosomes (Fig. 3D; Appendix Fig. S5d,e). We next addressed the recruitment of Cx43 to injured phagosomes containing *Salmonella* or effectene-coated latex beads (Kobayashi et al, 2010; Yoshida et al, 2017). Supporting our model, confocal images revealed that phagosomes containing *Salmonella* (Fig. 3e) or loaded with beads which mimic invading bacteria (Kobayashi et al, 2010) (Appendix Fig. S5f) present an accumulation of Cx43, which are also positive for Gal3. In addition, using human fibroblasts from patients with Niemann-Pick (NP) disorder, namely NPA1 and NPC1, which present loss of lysosomal membrane integrity (Appendix Fig. S5g,h), as previously reported (Kaufmann and Krise, 2008; Davis et al, 2021; Pascua-Maestro et al, 2020), we found an increased colocalization of Cx43 with Gal3 positive vesicles, when compared with control fibroblasts, in basal conditions, which is exacerbated by treatment with LLOMe (Appendix Fig. S5G–I). Interestingly, these findings were also followed by an increase in exocytosis (Appendix Fig. S5j,k). In contrast, Cx43 depletion in NPA1 and NPC1 fibroblasts resulted in

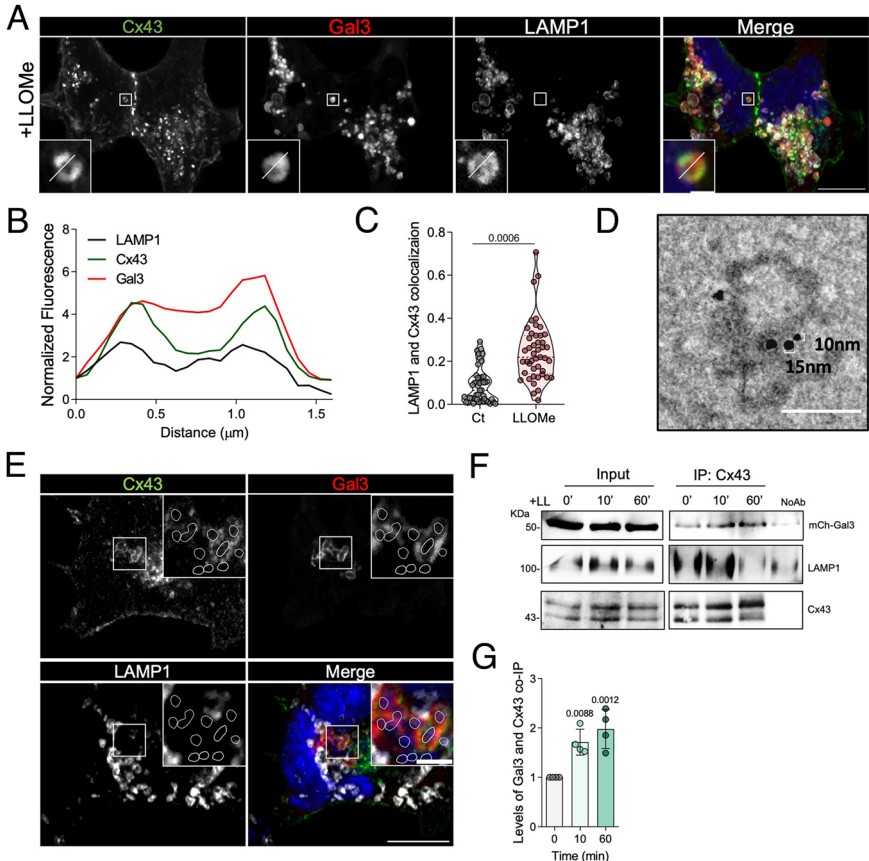

**Figure 3.   Cx43 is recruited to damaged lysosomes and interacts with Galectin-3.**

(**A**) Fluorescence images of GFP-Cx43, mCherry-Gal3 and immunostained LAMP1 in HEK293A LLOMe-treated cells during 60 min. Scale bar, 10 µm. The insets are enlargements of the boxed areas, 1 µm. (**B**) Fluorescence intensity profile of GFP-Cx43, mCherry-Gal3, and LAMP1 along the white line crossing the selected lysosome in the inset of (**A**). (**C**) Manders' correlation coefficients of fluorescence-intensity-based colocalization of GFP-Cx43 with LAMP1 calculated from confocal sections of cells treated as in (**A**). $n = 45$ cells, 15 cells per condition from three independent experiments. For each violin plot, the middle line denotes the median, the top and bottom lines indicate the 75th and 25th percentiles. *P* values were calculated by unpaired two-tailed t-test. (**D**) Transmission electron microscopy images of an enriched lysosomal pool immunogold stained for Cx43 (nanoparticle size of 10 nm) and LAMP1 (nanoparticles of 15 nm). Scale bar, 160 nm. (**E**) Fluorescence images of GFP-Cx43, mCherry-Gal3 and immunostained LAMP1 in HEK293A infected for 60 min with *Salmonella*. DAPI stains HEK293A nucleus and *Salmonella* DNA. White lines highlight the pathogens at the cropped images. Scale bar, 10 µm. The insets are enlargements of the boxed areas, 2 µm. (**F**) Immunoblot analysis of mCherry-Gal3 and LAMP1 following Cx43 immunoprecipitation. HEK293$^{Cx43+}$ cells were treated for 10 or 60 min with LLOMe (LL). (**G**) Quantification of mCherry-Gal3 co-immunoprecipitated with Cx43. Data are means ± SEM. $n = 4$. *P* values were calculated by one-way ANOVA with Dunnett's multiple comparisons test. Source data are available online for this figure.

a reduction of LAMP1 at cell surface (Appendix Fig. S5l). Of note, NPA1 and NPC1 fibroblasts present in basal conditions increased levels of Cx43 (Appendix Fig. S5m). Cx43 recruitment to damaged lysosomes was corroborated by co-immunoprecipitation, in which a redistribution of Cx43 to injured lysosomes was correlated with an increase in the interaction with Gal3 (Fig. 3F, G).

Next we sought to evaluate the impact of lysosomal damage on Cx43 turnover, by measuring its half-life in the presence of cycloheximide (CHX) with or without LLOMe or bafilomycin A1 (Baf) (Appendix Fig. S5n,o). Expectedly, in the presence of the lysosomal inhibitor Baf, degradation of Cx43 is prevented. However, only a slight reduction of Cx43 degradation was observed in cells incubated with LLOMe, suggesting that Cx43 redistribution to lysosomes, namely after lysosomal damage, plays a biological role and is not merely the final stage of the protein's life cycle. The data gathered up to this stage ascribes to Cx43 a new unanticipated role in the mechanisms of lysosomal membrane quality control.

## Cx43 is recruited to damaged lysosomes via its AP2 domain and endocytosis

Trafficking of newly synthesized transmembrane proteins from the trans-Golgi network to lysosomes can occur via a direct intracellular pathway, or indirectly through the plasma membrane (Luzio et al, 2007). To investigate the pathway used by Cx43 to reach damaged lysosomes, we started by quantifying the colocalization of Cx43 with LAMP1-positive vesicles in conditions of inhibited endocytosis resorting to genetic and pharmacological tools commonly used to modulate endocytic trafficking (Appendix Table S1). Chloropromazine (CPZ) and sucrose were used as clathrin-mediated endocytosis inhibitors, whereas filipin, β-methyl-cyclodextrin (MβC), nystatin and genistein were used to disturb caveolae-dependent endocytosis. Dynasore was used to inhibit dynamin-dependent processes. The obtained results show a reduction of Cx43 colocalization with lysosomes in cells simultaneously treated with LLOMe and the

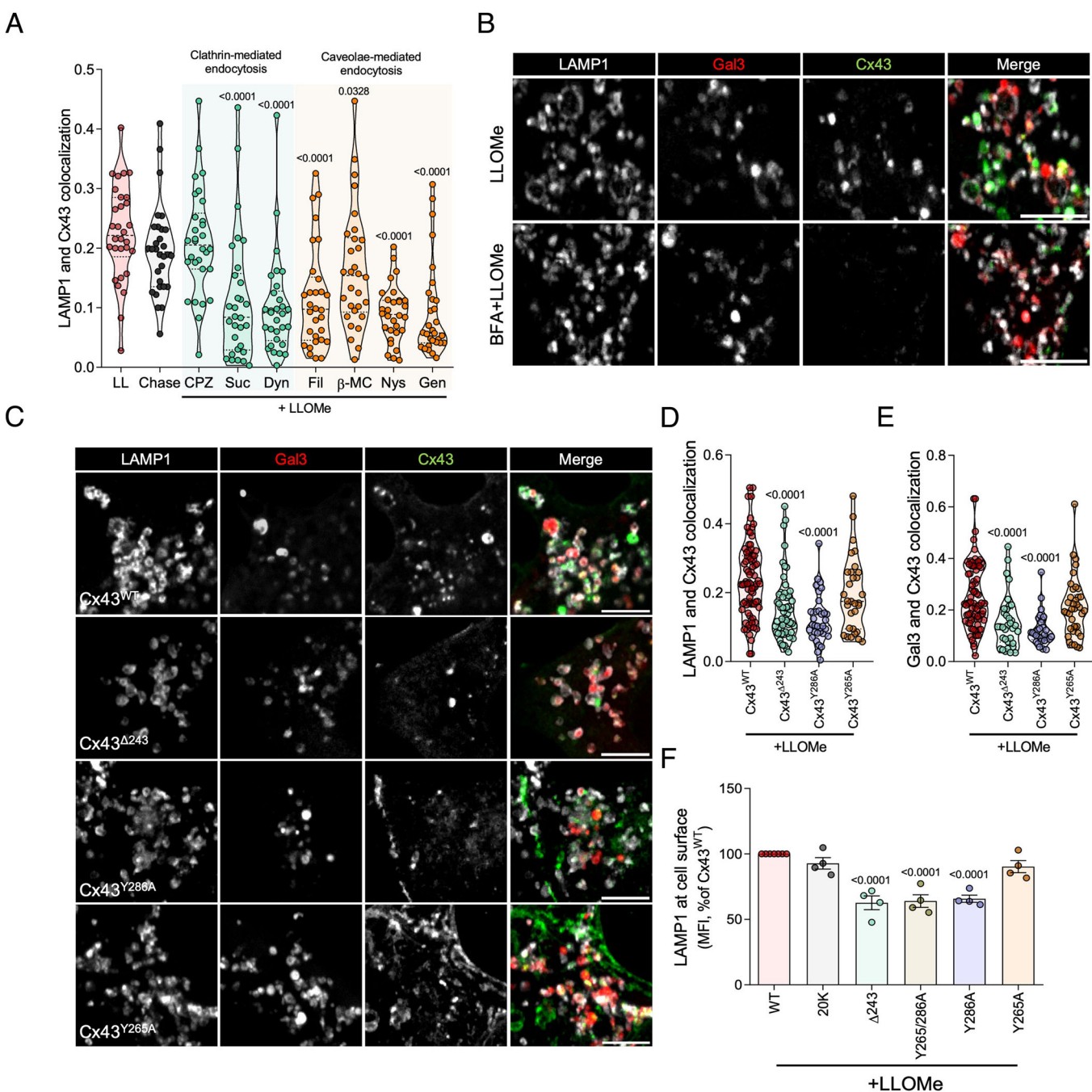

endocytic inhibitors, except for CPZ (Fig. 4A; Appendix Fig. S6a). Corroborating these findings, depletion of Epidermal Growth Factor Receptor Pathway Substrate 15 (Eps15) or overexpression of Rab5[Q79L], which induces Cx43 accumulation at the plasma membrane (Girão et al, 2009) or in early endosomes (Martins-Marques et al, 2020), respectively, decreased the colocalization of Cx43 and Gal3 (Appendix Fig. S6b–f). In addition, we performed a pulse-chase experiment in the presence of brefeldin A (BFA) (Fig. 4B), a drug that causes the fragmentation of the Golgi apparatus, disrupting the delivery of Cx43 to the plasma membrane (Smyth et al, 2012). Notably, Cx43 colocalization with LAMP1 (Appendix Fig. S6g) and Gal3-positive vesicles (Appendix Fig. S6h) decreased in the presence

of BFA (Fig. 4B). We confirmed these results using a mutant in which the -K-K-M-P (Lys-Lys-Met-Pro) ER retention motif was fused to the carboxylic terminal of Cx43 (Cx43[KKMP]), and by depletion of endoplasmic reticulum (ER) protein of 29 kDa (ERp29), which in both cases inhibit the biosynthesis and traffic of Cx43 to the plasma membrane (Das et al, 2009; Stornaiuolo et al, 2003). The results presented in Appendix Fig. S6i–m show a prominent reduction of Cx43 colocalization with Gal3 upon lysosomal damage when Cx43 transport to the cell surface is disrupted. Furthermore, colocalization levels were restored to basal levels after BFA washout, suggesting that the plasma membrane is the main source of Cx43 that is trafficked to damaged lysosomes.

**Figure 4. Cx43 is recruited to damaged lysosomes via endocytosis from the plasma membrane and is dependent on AP2 binding.**

(**A**) Manders' correlation coefficients of fluorescence-intensity-based colocalization of GFP-Cx43 with LAMP1 calculated from confocal sections of cells treated with LLOMe for 30 min followed by 120 min incubation with chlorpromazine (CPZ, 10 µM), sucrose (Suc, 250 mM), dynasore (Dyn, 80 µM), filipin (Fil, 5 µg/mL), β-methyl-cyclodextrin (β-MC, 1 mM), nystatin (Nys, 25 µM) and genistein (Gen, 200 µM) or fresh media (Chase). This latter condition was used as control for the reduced LLOMe internalization level in the presence of endocytic inhibitors. $n = 30$ cells per condition, 10 cells per condition from three independent experiments. For each violin plot, the middle line denotes the median, the top and bottom lines indicate the 75th and 25th percentiles. (**B**) Fluorescence confocal images of GFP-Cx43 and mCherry-Gal3 transiently transfected HEK293A cells treated with LLOMe for 60 min with or without a pre-treatment with Brefeldin A (BFA) for 60 min. After fixation, cells were immunostained for LAMP1. Scale bar, 5 µm. (**C**) Fluorescence confocal images of wild type (Cx43$^{WT}$) or mutant (Cx43$^{Δ243}$, Cx43$^{Y286A}$, Cx43$^{Y265A}$) GFP-Cx43, mCherry-Gal3 and immunostained LAMP1 in cells treated with LLOMe for 60 min. Scale bar, 5 µm. (**D, E**) Manders' correlation coefficients of fluorescence-intensity-based colocalization of GFP-Cx43 with LAMP1 (**D**) or mCherry-Gal3 (**E**) calculated from confocal sections of cells treated with LLOMe for 60 min with the same conditions mentioned in (**C**). $n = 30$–82 cells per condition, at least 10 cells per condition were analysed from three independent experiments. For each violin plot, the middle line denotes the median, the top and bottom lines indicate the 75th and 25th percentiles. (**F**) Quantification of the mean intensity of cell surface LAMP1 measured by flow cytometry in HEK293A cells expressing wild type Cx43 (Cx43$^{WT}$), Cx43-20k isoform or mutant Cx43 (Cx43$^{Δ243}$, Cx43$^{Y265/286A}$, Cx43$^{Y286A}$, Cx43$^{Y265A}$) following LLOMe treatment for 60 min. Data are means ± SEM. $n = 4$. For all the panels, P values were calculated by unpaired one-way ANOVA with Dunnett's multiple comparisons test. Source data are available online for this figure.

To further elucidate the mechanism involved in Cx43 recruitment to damaged lysosomes, we evaluated the impact of Cx43 mutations, associated with trafficking defects, in the colocalization with LAMP1 and Gal3-positive lysosomes. We transiently transfected cells with a C-terminal truncated form of Cx43 (Cx43$^{Δ243}$) and two Cx43 constructs with mutations in their tyrosine-sorting signals (Cx43$^{Y286A}$ and Cx43$^{Y265A}$), known to be important for Cx43 endocytosis through the binding to the clathrin adaptor-2 (AP-2) (Fong et al, 2013). In accordance with the data shown above, the endocytosis-defective mutants Cx43$^{Δ243}$ and Cx43$^{Y286A}$ presented a decreased colocalization with both LAMP1 and Gal3-positive lysosomes when compared with the wild type form of Cx43 (Fig. 4C–E). However, the recruitment of Cx43$^{Y265A}$ to damaged vesicles was not significantly affected. Similarly, deletion of the C-terminus or mutation of Y286 also impaired the ability of Cx43 to potentiate lysosome secretion triggered by lysosomal injury (Fig. 4F). Interestingly, overexpression of a naturally occurring truncated isoform of Cx43 (GJA1-20k), consisting of part of the final transmembrane domain and the full C-terminal of the protein, was enough to increase LAMP1 levels at the plasma membrane in response to LLOMe, similar to the wild type protein (Fig. 4F). Of note, we also observed that GJA1-20k was recruited to damaged lysosomes to similar levels as the wild type form of the protein (Appendix Fig. S6n–p). Thus, our findings strongly suggest that in order to modulate the exocytosis of lysosomes in response to damage, Cx43 is redistributed from the plasma membrane to lysosomes via AP2-dependent endocytosis.

## Cx43 potentiates exocytosis of damaged lysosomes through actin cytoskeleton modulation

An increase in the local Ca$^{2+}$ concentration near the cell surface is a key event that triggers the fusion of lysosomes with the plasma membrane (Reddy et al, 2001). Considering the role of Cx43 as an ionic channel, we sought to evaluate whether the effect of Cx43 in lysosome secretion could be related to its channel activity. To address this question, cells were transfected with Cx43 mutants that form constitutively open (Cx43$^{S368A}$) or closed channels (Cx43$^{S368D}$), after which channel activity was assessed using lucifer yellow assay (Appendix Fig. S7a). As depicted in Fig. 5A, following LLOMe treatment, we observed no differences in LAMP1 intensity

at the cell surface in both mutants when compared with wild-type Cx43. To further confirm these results, we used the connexin mimetic peptide gap26, which blocks Cx43 channel function (Appendix Fig. S7b) (Evans and Leybaert, 2009). Again, the lack of any impact of gap26 strengthens the hypothesis that Cx43 channel function is not required to promote lysosomal exocytosis (Fig. 5B). Moreover, we provided evidence that this Cx43-related lysosome exocytosis is independent of Ca$^{2+}$, since treatment with the Ca$^{2+}$ chelator BAPTA did not revert the Cx43-mediated increase of LAMP1 at the cell surface after lysosomal rupture. In addition, ionomycin/Ca$^{2+}$ treatment did not potentiate the LLOMe-induced increase in LAMP1 cell surface levels induced by Cx43 expression (Fig. 5B). Previous studies have described a Ca$^{2+}$-independent exocytosis process for synaptic vesicles induced by a short burst of hypertonic sucrose, a process relying on actin polymerization (Orlando et al, 2019). Here, we found that this treatment also exacerbated the effect of LLOMe on lysosome secretion in both cell lines. Importantly, when LLOMe-treated cells were incubated with the actin polymerization inhibitor (cytochalasin D, Cyto D), a microtubule polymerization inhibitor (nocodazole), or the actin stabilizing agent (jasplakinolide), the increase of LAMP1 at the cell surface in HEK293$^{Cx43+}$ cells was abolished, presenting similar values to HEK293A cells (Fig. 5B; Appendix Fig. S7c,d). These data support a model in which the Cx43-dependent exocytosis mechanism relies on the ability of Cx43 to reorganize the actin cytoskeleton. To further address this issue, we then investigated the impact of Cx43 on the actin network in response to lysosomal damage. Notably, cells expressing Cx43 display changes in actin organization, with an increase in small actin branched fragments similar to actin aster structures (Fritzsche et al, 2017), which were more evident after LLOMe treatment (Fig. 5C). Strikingly, we also observed that Cx43 expression potentiated bleb formation in LLOMe treated cells (Appendix Fig. S7e,f), reinforcing a potential effect of Cx43 on the organization of cortical actin. Corroborating this finding, a significant alteration in actin organization was detected in SUM$^{Cx43KO}$ cells (Appendix Fig. S7g). To further investigate this phenotype, we applied two different image analysis plugins to extract and analyse the actin cytoskeleton network (Fig. 5D,E; Appendix Fig. S7h–l). In basal conditions HEK293$^{Cx43+}$ cells presented an increased number of total actin branches (Fig. 5D) and multiple branches (Fig. 5E), encompassing triple and quadruple branching points, when compared to parental cells.

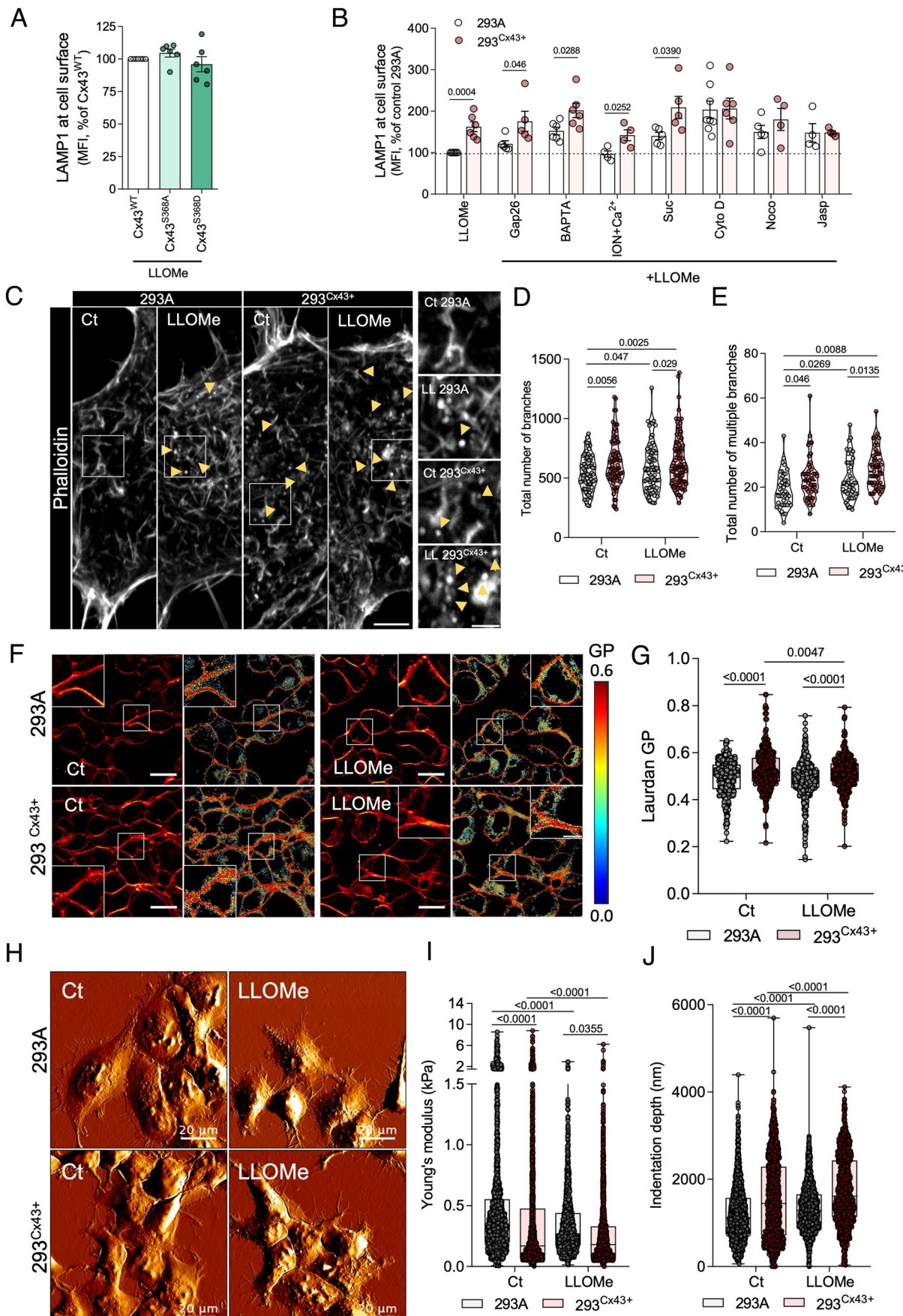

**Figure 5. Cx43 effect on exocytosis involves actin network dynamics.**

(A) Quantification of the mean intensity of cell surface LAMP1 by flow cytometry after LLOMe treatment for 60 min of HEK293A cells expressing wild type (Cx43$^{WT}$) or mutant (Cx43$^{S368A}$ and Cx43$^{S368D}$) Cx43. Data are means ± SEM. $n = 6$. (B) Quantification of the mean intensity of cell surface LAMP1 by flow cytometry in cells treated with LLOMe for 60 min in the presence of different compounds: Gap26 was added to cells 60 min before LLOMe incubation; BAPTA and ionomycin (Ion) and calcium (Ca$^{2+}$) were added 10 min before LLOMe treatment ended. Sucrose was added 90 s before LLOMe treatment ended; cytochalasin D, nocodazole and jasplakinolide (Jasp) were added 30 min before the end of LLOMe treatment. Data are means ± SEM. At least $n = 4$ for each condition. (C) Fluorescence images of HEK293A and HEK293$^{Cx43+}$ control and LLOMe treated cells stained for F-actin with phalloidin. Yellow arrows point to actin patches structures. Scale bar, 5 μm. (D, E) Quantification of total number of branches (D) and multiple branches (E) from confocal section images from the conditions represented in (C). At least 45 cells were analysed, at least 15 cells per condition in each independent experiment. Data are means ± SEM. $n = 3$. (F) Two-photon excitation microscopy of HEK293A and HEK293$^{Cx43+}$ control and LLOMe treated cells labelled with Laurdan. Images from the left panel correspond to the sum of equatorial z-section images taken with emission set at 400–460 nm and 470–530 nm. Generalized polarization (GP) images (right) were obtained by applying the GP function (methods section) to the images from both channels, pixel by pixel. GP scale to pseudo colour the intensity image is shown on the right. Scale bar, 10 μm. (G) GP values obtained for the plasma membrane after incubation with Laurdan. At least 400 different cells were imaged per experimental condition, during 3 independent experiments. (H) Representative peak force error mode of atomic force microscopy (AFM) images of HEK293A and HEK293$^{Cx43+}$ control and LLOMe treated cells for 60 min. All the images present a fixed scale of 500 mV. (I) Young's modulus data for each cell condition represented in (H). At least 400 different cells were analysed per experimental condition, 3 independent experiments. Data are means ± SEM. (J) Penetration depth of the AFM tip on HEK293A and HEK293$^{Cx43+}$ control and LLOMe treated cells for 60 min. At least 400 different cells were analysed per experimental condition, 3 independent experiments. Data are means ± SEM. Approximately 440 cells were analysed per condition, with 5 force-distance curves per each cell. Three independent experiments were performed. $P$ values were calculated by one-way ANOVA with Tukey's multiple comparisons test in (A) and two-way ANOVA with Sidak's multiple comparisons test (B, D, E, G, I, J). For each violin plot, the middle line denotes the median, the top and bottom lines indicate the 75th and 25th percentiles. In the Box and whiskers graphs, minimum and maximum whiskers, the black line represents the median. Source data are available online for this figure.

After lysosomal damage, an increase in actin branching was observed, with this effect being more evident in HEK293$^{Cx43+}$ cells. In addition, LLOMe treatment affected actin density (Appendix Fig. S7h) and intensity of distribution (Appendix Fig. S7i), but did not influence the parallel alignment, angle and actin branch length (Appendix Fig. S7j–l). Interestingly, when cells were transfected with Cx43 constructs mutated in endocytic domains (Cx43$^{Y286A}$ or Cx43$^{Y265A+Y286A}$), which hamper LLOMe-mediated lysosomal secretion (Fig. 4F), these observed actin patches structures were reduced (Appendix Fig. S7m). Moreover, depletion of the previously described actin-binding motif in the C-terminal of Cx43 (Cx43$^{Δ373}$) (preprint: Baum et al, 2022), led to a decrease in Cx43 and actin colocalization near galectin-positive lysosomes (Appendix Fig. S7n) and LAMP1 at the cell surface (Appendix Fig. S7o,p) upon LLOMe treatment, when compared to cells overexpressing Cx43$^{WT}$.

It is well established that cortical actin organization is vital to modulate plasma membrane architecture and dynamics (Fritzsche et al, 2017). Therefore, we investigated the impact of Cx43 on the overall organization of the plasma membrane, during the secretion of injured lysosomes. To address this question, we used two-photon fluorescence microscopy with the environment-sensitive fluorescent probe Laurdan to calculate the generalized polarization (GP) value, an indicator of plasma membrane fluidity (Fig. 5F,G). Our results show that expression of Cx43 caused a membrane fluidity decrease, demonstrated by the higher GP values. Interestingly, an increase in plasma membrane fluidity was observed in Cx43 expressing cells after treatment with LLOMe. We also assessed the impact of Cx43 and lysosomal injury in cell stiffness, quantified as Young's modulus and indentation depth by atomic force microscopy (AFM) (Appendix Fig. S8a). Results depicted in Fig. 5H and Appendix Fig. S8b reveal morphological alterations between the two cell lines and after LLOMe treatment, with Cx43 expression causing an increase in cellular height whereas LLOMe induced a reduction. In addition, HEK293$^{Cx43+}$ cells presented lower Young's modulus values and higher tip penetration depth when compared with control cells, indicating higher cell elasticity (Fig. 5H–J). After lysosomal damage, a decrease in Young's

modulus and an increase in indentation values was verified, with cells expressing Cx43 presenting higher variation in cell elasticity. Overall, these results suggest that lysosomal exocytosis potentiated by Cx43 increases membrane fluidity and the ability of the cell to deform upon lysosomal damage.

## Cx43 modulates the actin network through Arp2 activity

The Arp2/3 complex is responsible for nucleating branches in actin filaments and transforming actin vortices to asters in order to produce new F-actin networks and to facilitate several cellular processes, including cell membrane adherence (Fritzsche et al, 2017; Colin-York et al, 2019) and fusion (Billault-Chaumartin and Martin, 2019). This led us to investigate whether Cx43-induced actin remodelling involved Arp2 activity. We first investigated the impact of lysosomal damage upon Arp2 distribution. Fluorescence confocal images revealed that, following LLOMe treatment, GFP-Arp2 exhibited a puncta distribution, a phenotype that was potentiated by Cx43 expression (Fig. 6A–C and controls in Appendix Fig. S9a). Our images further revealed a distribution of Arp2 near actin filaments in the proximity of LAMP1 (Fig. 6A), Gal3 and Cx43 (Appendix Fig. S9b) positive vesicles. Importantly, this increase in Arp2 recruitment to actin filaments was also found around engulfed *Salmonella* in infected-HEK293$^{Cx43+}$ (Fig. 6D), as well as in lysosomes from NPA1 and NPC1 fibroblasts (Appendix Fig. S10c). NPA1 and NPC1, which presented a branched and patched actin cytoskeleton, also demonstrated Cx43 recruitment to actin positive lysosomes near the plasma membrane (Appendix Fig. S10d). Of note, the Arp2-punctate distribution was exacerbated when actin filaments were disrupted by Cyto D, with this effect being more pronounced in cells pre-treated with LLOMe (Appendix Fig. S9c). Considering these findings and the overlapping of Cx43 with actin (Movie EV2) and lysosomes after damage induction (Appendix Fig. S10a,b), we next investigated whether Cx43 could form complexes involving Arp2. Fluorescence confocal microscopy assays clearly showed that LLOMe treatment resulted in higher levels of colocalization between Cx43 and Arp2

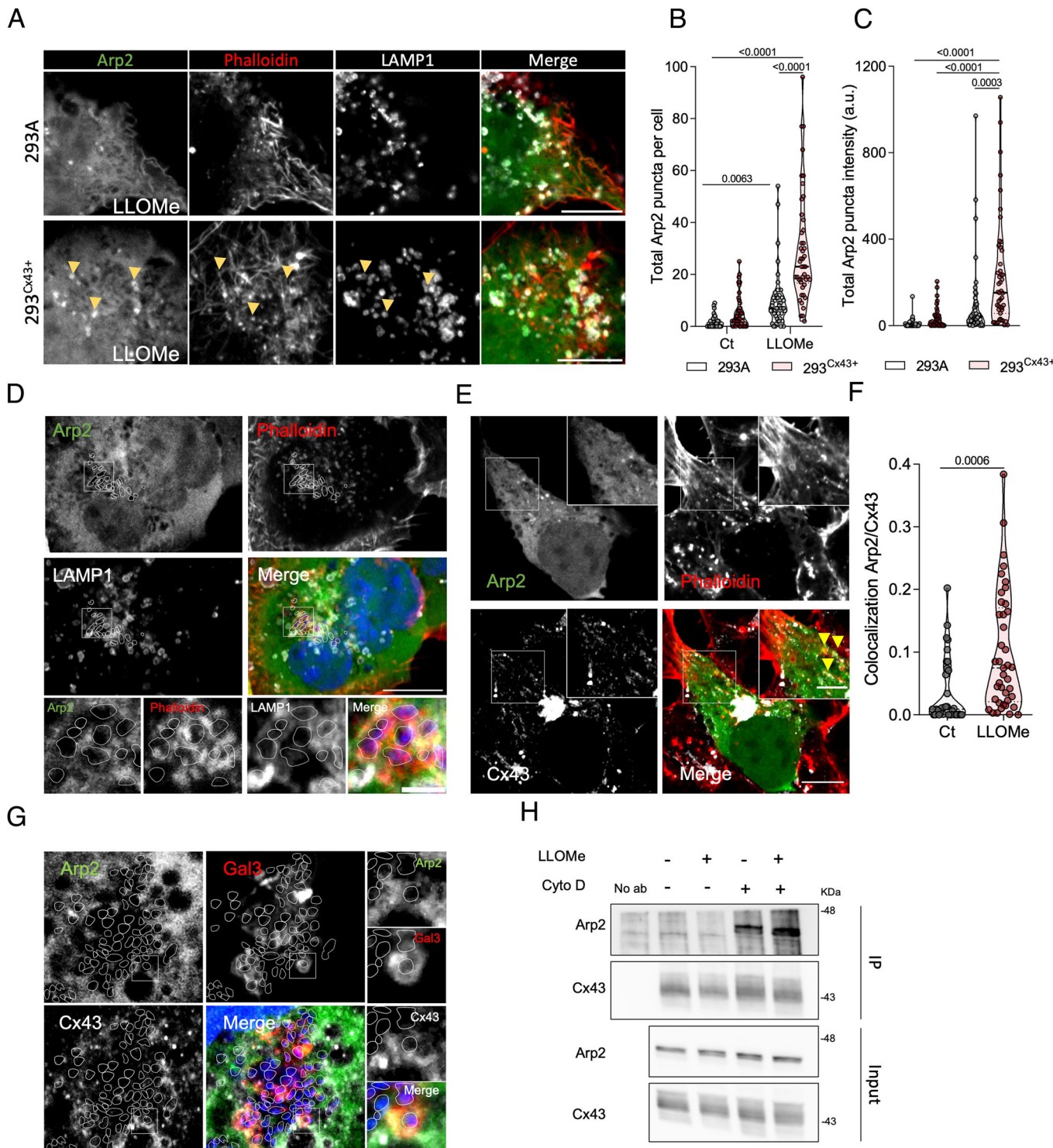

(Fig. 6E,F; Appendix Fig. S10e). Also, this colocalization between Cx43 and Arp2 was observed in actin filaments surrounding *Salmonella* (Appendix Fig. S10f) and in Gal3-positive lysosomes containing these pathogens (Fig. 6G). The existence of a complex including Arp2 and Cx43 was corroborated by the results obtained by co-precipitation assays, indicating that these two proteins are present in the same structures (Fig. 6H). Interestingly, we found that Arp2 co-immunoprecipitation with Cx43 was potentiated when the actin network was disrupted with Cyto D (Fig. 6H; Appendix Fig. S10g). Taken together, these results suggest that Cx43-dependent modulation of the actin network is associated with Arp2-dependent actin nucleation.

**Figure 6.  Cx43 interacts with Arp2 and enhances Arp2-puncta distribution following lysosomal injury.**

(A) Confocal fluorescence images of HEK293A and HEK293$^{Cx43+}$ expressing GFP-Arp2. Following LLOMe treatment for 60 min, cells were immunostained for LAMP1 and actin was labelled with phalloidin. Yellow arrows point to Arp2 puncta colocalizing with actin and LAMP1 vesicles. (B, C) Total number of Arp2 puncta (B) and total of Arp2 intensity (C) per cells from conditions mentioned in (A) Data are means ± SEM. At least $n = 45$ cells were analysed per condition, with a minimum of 15 per each of the three independent experiments. (D) Fluorescence images of HEK293$^{Cx43+}$ cells transfected with GFP-Arp2, immunostained for LAMP1 and actin labelled with phalloidin after infection for 60 min with *Salmonella*. In the merged image, DAPI stains HEK293$^{Cx43+}$ nucleus and *Salmonella* DNA. White lines highlight the pathogens defined by DAPI staining. (E) Confocal fluorescence images of HEK293$^{Cx43+}$ cells transiently transfected with GFP-Arp2 and treated with or without LLOMe during 60 min. Cx43 was detected by immunofluorescence labelling and actin by phalloidin staining. Yellow arrows point to Arp2 puncta colocalizing with actin and Cx43. (F) Manders' correlation coefficients of fluorescence-intensity-based colocalization of GFP-Arp2 with Cx43 calculated from (E). Data are means ± SEM. At least $n = 34$ cells were analysed per condition, with a minimum of 10 per each of the three independent experiments. (G) Fluorescence images of HEK293$^{Cx43+}$ cells transfected with GFP-Arp2 and mCherry-Gal3, immunostained for Cx43. Cells were infected for 60 min with *Salmonella*. In the merged image, DAPI stains the nucleus of HEK293$^{Cx43+}$ and *Salmonella*. White lines indicate the pathogen defined by DAPI staining. (H) Representative immunoblot image of Arp2 following Cx43 immunoprecipitation from HEK293$^{Cx43+}$ cells treated with or without LLOMe for 60 min, and during the last 30 min, with cytochalasin D (Cyto D). (A, D, E, G) Scale bar 10 μm. The insets are enlargements of the boxed areas, 2 μm. $P$ values were calculated by two-way ANOVA with Sidak's multiple comparisons test for (B, C) and t test for (F). For each violin plot, the middle line denotes the median, the top and bottom lines indicate the 75th and 25th percentiles. Source data are available online for this figure.

## Cx43 modulates exocytosis through Arp2

Arp2/3-mediated actin branch formation is essential for large secretory vesicles to access the plasma membrane and membrane fusion (Carisey et al, 2018). Therefore, we proceeded to unravel the role of Arp2/3 in Cx43-mediated actin remodelling. Remarkably, we found that Arp2 silencing hampered the actin changes induced by Cx43 expression, namely the patch structures (Fig. 7A) and the number of branches and multiple branched actin filaments (Fig. 7B,C; Appendix Fig. S11a). We next investigated the impact of Arp2 activity on Cx43-mediated exocytosis by measuring LAMP1 cell surface levels and the release of lysosomal Cath B. In agreement, downregulation of Arp2 in LLOMe-treated HEK293$^{Cx43+}$ cells partially hindered the increase in LAMP1 intensity (Fig. 7D,E) and Cath B secretion (Fig. 7F), when compared with control cells. We further assessed the importance of Arp2/3 activity in Cx43-mediated lysosomal secretion using the Arp2 chemical inhibitors CK666 and CK869. Our data showed that both inhibitors decreased the levels of LAMP1 at the cell surface in HEK293$^{Cx43+}$ cells (Appendix Fig. S11b,c) and the secretion of Cx43 and LAMP1 in extracellular vesicles, upon LLOMe treatment (Appendix Fig. S11d).

Strikingly, our microscopy data revealed that the combination of Cx43 expression, LLOMe treatment and Arp2 downregulation (Fig. 7G) or inhibition (Appendix Fig. S11e) led to an increase in the formation of plasma membrane blebs. These large structures were also frequently found in TEM images of HEK293$^{Cx43+}$ treated with LLOMe under Arp2 inhibition (Appendix Fig. S11f). We observed that LAMP1 and Cx43 could both be found mainly at the plasma membrane-anchored base of blebs, which was more evident when bleb formation was potentiated in Arp2-depleted HEK293$^{Cx43+}$ cells (Fig. 7G) or treated with Arp2 inhibitors (Appendix Fig. S12a). Thus, although Arp2 inhibition was previously shown to induce bleb formation in basal conditions (Obeidy et al, 2020; Bergert et al, 2012), here we showed an exacerbated blebbing phenotype only upon lysosome damage.

Membrane blebbing has been associated with local increases in Ca$^{2+}$ concentration (Nieminen et al, 1988). Considering the rise in Ca$^{2+}$ leakage from damaged lysosomes, we investigated whether the observed phenotype of membrane blebbing could be related with the accumulation of unfused damaged lysosomes near the plasma membrane upon Arp2 depletion (Appendix Fig. S12b). Our data show a prominent reduction in the number and size of these blebs

when intracellular levels of Ca$^{2+}$ were reduced by BAPTA treatment (Appendix Fig. S12c), suggesting that bleb formation following LLOMe treatment and Arp2 inhibition in HE293$^{Cx43+}$ cells is, at least partially, due to a rise in local Ca$^{2+}$ concentration, likely resulting from the accumulation of leaky lysosomes near the cell border. Altogether these results strongly suggest that Cx43 promotes exocytosis induced by lysosomal damage through an Arp2/3-dependent actin remodelling process.

# Discussion

The accumulation of dysfunctional lysosomes with compromised membrane integrity has been implicated in various pathologies (Medina et al, 2011; Palmieri et al, 2017; Samie and Xu, 2014; Spampanato et al, 2013). Thus, strategies to promote the repair and/or enhance the clearance of these lysosomes have been proposed as a potential therapeutic approach for several disorders, including neurodegenerative disorders, such as Batten (Palmieri et al, 2017), Huntington (Sardiello et al, 2009) and Parkinson's disease (Decressac et al, 2013). Some studies have implicated exocytosis as an additional level of lysosomal quality control (Burbidge et al, 2022; Xu et al, 2021; Medina et al, 2011); however, the molecular cues mediating the interplay between this mechanism and lysosomal repair/elimination remain largely elusive. In the present study, we bring new insights on the mechanisms of lysosomal exocytosis as part of a cellular response to lysosomal damage when repair and degradation are inhibited. Indeed, we identified Cx43 as a new player involved in lysosomal exocytosis. Following lysosomal membrane disruption, we found that Cx43 is recruited to damaged lysosomes promoting their fusion with the plasma membrane, with subsequent secretion of lysosomal content and the outward budding of the cell surface release of microvesicles to the extracellular milieu. In our model, the increase in predocked Cx43-enriched damaged lysosomes close to cortical actin activates the remodelling of the cytoskeleton in order to facilitate their secretion, in a process regulated by the Arp2/3 complex (Fig. 7H). Similar to what has been described for the exocytosis of synaptic vesicles (Orlando et al, 2019), we found that exocytosis of damaged lysosomes is Ca$^{2+}$-independent and controlled by the actin network. Interestingly, an interactome study of injured lysosomes, revealed an interaction between Gal9, another galectin known to be recruited to damaged lysosomes, and VAMP7 (Jia et al, 2020a), a SNARE protein involved in the fusion of lysosomes with the plasma membrane.

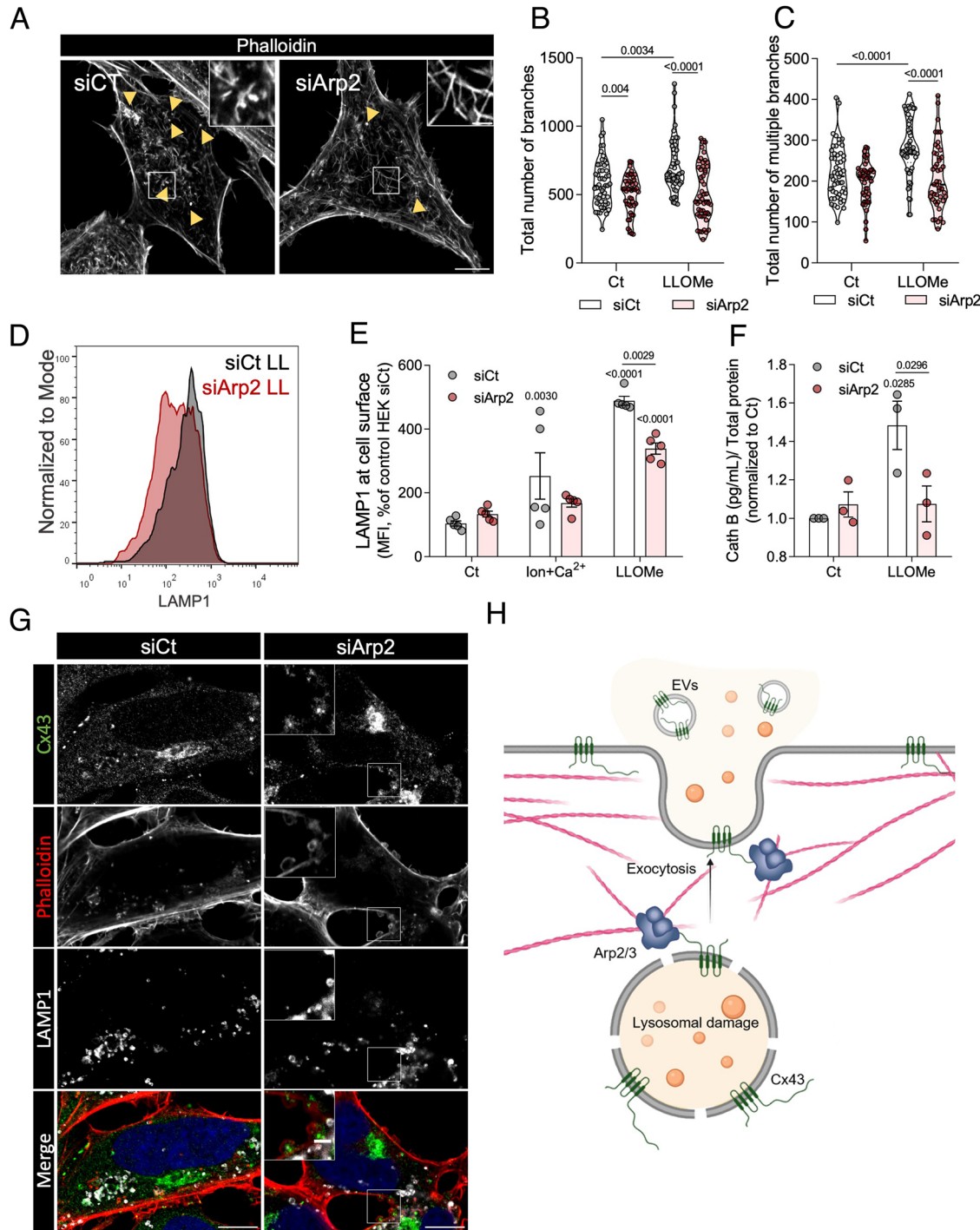

Nevertheless, how fusion machinery associated with secretory lysosomes, such as VAMP7 and CD63/LAMP3 (Blott and Griffiths, 2002; Li et al, 2008; Verderio et al, 2012; Domingues et al, 2023), is specifically recruited to damaged lysosomes is still unclear. Interestingly, a recent report showed increased cellular secretion in response to α-synuclein-induced lysosomal damage through a mechanism that relies on the autophagic secretory pathway, with a direct contribution of Gal3, Trim16 (tripartite motif containing 16) and ATG16L1 (autophagy-related 16 like1) (Burbidge et al, 2022).

Our study provides new evidence to an emerging field of research devoted to non-canonical functions of Cx43, which goes beyond gap junction-mediated intercellular communication (Martins-Marques et al, 2019). Previous studies from our group have shown that extracellular vesicles are enriched in Cx43 that participates in the selective sorting of microRNA into extracellular vesicles (Martins-Marques et al, 2022) and their release into recipient cells (Martins-Marques et al, 2016). Our present findings demonstrate that Cx43 recruited to damaged lysosomes serves to

◄

**Figure 7. Cx43-mediated actin network remodelling and exocytosis is dependent of Arp2.**

(A) Confocal fluorescence images of phalloidin-staining of F-actin from HEK293$^{Cx43+}$ control (siCt) or Arp2-silenced (siArp2) cells treated with LLOMe for 60 min. Yellow arrows point to actin patches structures. Scale bar, 10 μm. (B, C) Quantification of the total number of branches (B) and multiple branches (C) from confocal section images of HEK293$^{Cx43+}$ control or silenced to Arp2, treated with or without LLOMe. At least 45 cells were analysed, 15 cells per condition in each independent experiment. Data are means ± SEM. $n = 3$. For each violin plot, the middle line denotes the median, the top and bottom lines indicate the 75th and 25th percentiles. (D) Representative histogram of the mean intensity of cell surface LAMP1 measured by flow cytometry in HEK293$^{Cx43+}$ control or silenced-Arp2 cells treated with LLOMe (LL). (E) Quantification of the mean intensity of cell suface LAMP1 measured by flow cytometry in the same condition mentioned in (B, C). Ionomycin (Ion) and calcium (Ca$^{2+}$) were used as a positive control of lysosomal exocytosis. Data are means ± SEM. $n = 5$. (F) Cathepsin B (Cath B) release by HEK293$^{Cx43+}$ cells from the same conditions as in (B, C). Data are means ± SEM. $n = 3$. (G) Confocal fluorescence images of HEK293$^{Cx43+}$ cells silenced for Arp2 (siArp2) and treated with LLOMe for 60 min. Control cells were transfected with scramble siRNA (siCt). Cx43 and LAMP1 were detected by immunofluorescence labelling and actin by phalloidin staining. Scale bar, 10 μm. The insets are enlargements of the boxed areas, 2 μm. $P$ values were calculated by two-way ANOVA with Sidak's multiple comparisons test. (H) Working model of Cx43-mediated exocytosis as a strategy to eliminate damaged lysosomes. In conditions of defective lysosomal membrane repair and lysophagy, exocytosis emerges as an alternative strategy to eliminate damaged lysosomes. Following damage, Cx43 is internalized from the plasma membrane and relocated to damaged lysosomes. With an apparent effect on cytoskeleton organization, Cx43 increases the exocytosis capacity of the cells after lysosomal damage, promoting the release of lysosomal enzymes and extracellular vesicles (EVs) enriched in Cx43. Moreover, upon lysosomal damage, actin becomes branched and forms patch structures, a mechanism promoted by Cx43 and Arp2 interaction. $P$ values were calculated by two-way ANOVA with Sidak's multiple comparisons test for (B, C, E) and two-way ANOVA with Uncorrected Fisher's LSD (F). Source data are available online for this figure.

fuel the actin cytoskeleton rearrangement that facilitates exocytosis. Indeed, actin has been shown to present several overlapping functions during the tethering and fusion of vesicles with the plasma membrane, working as a barrier, a guide to the docking and fusion sites, and as a motor for fusion pore formation and cargo expulsion [reviewed in (Papadopulos, 2017)]. Nevertheless, the molecular players involved in these different actin functions during secretion remain unclear. Curiously, the GJA1-20K isoform was shown to stabilize the actin microtubule network, contributing to the delivery of Cx43 to the plasma membrane (Basheer et al, 2017). This isoform was also implicated in the anterograde transport of mitochondria (Fu et al, 2017), as well as in the trafficking of cardiac channels and junctional proteins to cell-cell borders (Agullo-Pascual et al, 2014; Jansen et al, 2012; Lübkemeier et al, 2013). These observations are in line with our data showing that upon lysosomal damage, Cx43 expression increases the pool of lysosomes near the plasma membrane. Despite all this evidence strongly suggesting a role for Cx43 in cytoskeleton modulation, a direct interaction between Cx43 and actin has never been reported. Previous studies demonstrated that Cx43 interacts with F-actin indirectly through the binding of Zonula occludens-1 (ZO-1) to the last four amino acid residues of the C-terminus (Toyofuku et al, 1998). Notably, recent data identified an RPxxxEL actin binding motif in the last nine amino acids of the C-terminus of Cx43-20K isoform (preprint: Baum et al, 2022). In accordance with our findings, the authors observed thickened actin filaments and an actin puncta phenotype in cells with higher levels of this Cx43 (preprint: Baum et al, 2022). However, until now the cellular implications of this modified actin network were not unravelled.

In the present study, we report that lysosomal damage was able to increase plasma membrane fluidity and decrease cell stiffness, a process further exacerbated by Cx43 expression. Removal of cortical actin, during the early events of secretion, was previously shown to induce a quick loss of tension at the incision point allowing the movement of tethered vesicles toward the plasma membrane (Papadopulos et al, 2015). This process was accomplished by a redistribution of apical phosphatidylinositol-4,5-bisphosphate (PI(4,5)P$_2$) to the vesicle membranes (Wen et al, 2011). Although, not tackled in the context of actin-mediating membrane dynamics, PI(4,5)P$_2$ was found to form a complex and regulate Cx43 channel function (Van Zeijl et al, 2007). Considering

the observed membrane fluidity reduction induced by Cx43, usually associated with lowered membrane tension, it is plausible that this effect is related to an increased ability of cells to form the fusion pore through a process that can involve interaction with PI(4,5)P$_2$. Accordingly, in conditions of cortical actin disruption, using actin polymerization inhibitors such as Cyto D, which induce a decrease in membrane tension (Lieber et al, 2013), fluidity (Kucik et al, 1996), and stiffness (Rotsch and Radmacher, 2000), a transient increase in secretion was reported (Wollman and Meyer, 2012). This suggests that rapid changes in cortical actin architecture confer the ability of vesicles to pass through this barrier, with consequent implications in plasma membrane dynamics. In addition, during later stages of secretory vesicle expulsion, namely docking and fusion, actin remodelling around the vesicles was found to rely on the interaction of PI(4,5)P$_2$ with the Arp2/3-Cdc42-N-WASP complex (Dong et al, 2010). In *Drosophila* cells, Arp2 or Arp3 depletion abolished the cargo expulsion and the membrane integration of the docked vesicles with the plasma membrane (Tran et al, 2015). These findings point to an important role of Arp2/3 during the coating of the cargo for release originating actin branched structures from linear filaments followed by the recruitment of the non-muscle myosin II (Papadopulos, 2017; Tran et al, 2015). Our results support a role for the Arp2 complex in exocytosis induced by lysosomal damage, and ascribe to Cx43 a new function during this coating process, through a mechanism that is likely related with the Arp2/3-Cdc42-N-WASP complex and subsequent fusion of membranes. Thus, we envision a model in which Cx43 interaction with the Arp2-complex regulates the actin-mediated secretion of damaged lysosomes.

By proposing Cx43 as a new druggable target, this study can open new avenues for the treatment of diseases associated with loss of lysosomal activity and membrane repair capacity, namely in lysosomal storage, infection, neurodegeneration and aging-related disorders, where exocytosis enhancement could ameliorate the pathophysiological conditions. In summary, our study highlights exocytosis as an important mechanism involved in the cellular response to lysosomal damage as an alternative to repair and degradation. We also provide strong evidence of a new role for the gap junction protein Cx43 in membrane quality control, given its ability to traffic to the membranes of injured lysosomes and,

through an Arp2-dependent process, regulate the actin network and promote exocytosis.

# Methods

## Chemical reagents

Chemical reagents were used at the following concentrations, unless indicated otherwise: 0.5 mM L-leucyl-L-leucine O-methyl ester (LLOMe, Bachem, 4000725); 5 μM ionomycin (Sigma-Aldrich, I0634); 4 mM calcium chloride (CaCl$_2$; MERCK, 102382); 30 μM propidium iodide (PI, Sigma-Aldrich, P4170); 10 μM BAPTA-AM (BAPTA, Cayman Chemical, 15551); 5 mM 3-methyladenine (3MA, Sigma-Aldrich, M9281); 5 μg/mL brefeldin A (BFA, Santa Cruz Biotechnology, sc-200861); sulfosuccinimidyl-2-[biotinamido]ethyl-1,3-dithiopropionate (sulfo-NHS-SS-biotin, ApexBio, A8005); 100 μM bafilomycin A1 (Baf, Santa Cruz Biotechnology, sc-201550); 10 μM chlorpromazine hydrochloride (CPZ, Sigma-Aldrich, C8138), 0.4 M or 250 mM sucrose (Suc, Panreac, 131621.1211); 80 μM dynasore (Dyn, Sigma-Aldrich, D7693); 5 μg/mL filipin (Fil, Sigma-Aldrich, F4767), 1 mM methyl-β-cyclodextrin (β-MC, Sigma-Aldrich, C4555); 25 μM nystatin (Nys, Sigma-Aldrich, N4014); 200 μM genistein (Gen, Sigma-Aldrich, G6649); 5 μg/mL cytochalasin D (Cyto D, Sigma-Aldrich, C8273); 2 μg/mL nocodazole (Noco, Sigma-Aldrich, M1404); 100 nM jasplakinolide (Jasp, EMD Millipore, 420127-50UG); phalloidin-TRITC (Sigma-Aldrich, P1951), 6-lauryl-2-dimethylamino-naphtalene (Laurdan, Thermo Fisher Scientific, D250). The synthetic peptide Gap26 (biotin-KQIEIKKFK) was obtained from CASLO ApS and used at 0.25 mg/mL. Protease inhibitor cocktail (Roche, 11873580001), Neutravidin beads (Pierce, 29200), blasticidin (Sigma-Aldrich, 15205), Mowiol 4-88 (Sigma-Aldrich, 81381), Lipofectamine™ 2000 (Thermo Fisher Scientific, 11668019), human cathepsin (Cath) B ELISA Kit (Abcam, ab119584), Protein G sepharose beads (GE Healthcare, 17061801), Lysosome Isolation kit (Sigma-Aldrich, LYSIO1), Osmium tetroxide solution (Sigma, 75632), uranyl acetate (Merck, M1775); 10 μM BAPTA-AM (BAPTA, Cayman Chemical, 15551); 5 mM 3-Methyladenine (3MA, Thermo Fisher Scientific; L7528); 250 μg/mL Fluorescein isothiocyanate-dextran (FITC-Dextran; Merck, 60842-46-8) 50 μg/mL Alexa Fluor 657-Dextran, 10000 MW (Thermo Fisher Scientific, D22914); DQ-BSA (Thermo fisher Scientific, D12051); 7 μM CK869 (Sigma-Aldrich, C9124-5MG); 20 μM CK666 (Sigma-Aldrich, SML0006-5MG), 5 mmol/L cycloheximide (CHX, Sigma-Aldrich, C-7698).

## Antibodies

The following antibodies were used: mouse monoclonal anti-LAMP1 [H4A3] (development studies hybridoma bank, AB_2296838; IB 1:1000; IF 1:100); mouse monoclonal anti-Galectin-3 [B2C10] (Santa Cruz, sc-32790; IB 1:1000); mouse monoclonal anti-N-cadherin [H-63] (Santa Cruz, sc-7939; IB: 1:1000); mouse monoclonal Alexa647-conjugated anti-LAMP1 [H4A3] (Biolegend, 328612; IF 1:200); mouse polyclonal anti-hnRNPA2B1 [B-7] (Santa Cruz, sc-374053; IB:1000); goat poly-clonal anti-Cx43 [3A9] (SICGEN, 634501, IB 1:5000; IF 1:200); mouse monoclonal anti-tubulin (Sigma-Aldrich, T6199; IB 1:1000); rabbit monoclonal anti-Arp2 (Atlas Antibodies, HPA008352; IB 1:1000; IF 1:100); Mouse monoclonal anti-Alix [1A12] (Santa Cruz, sc-53540; IB 1:1000); mouse monoclonal anti-Calnexin (SICGEN, AB0041; IB 1:5000); rabbit polyclonal anti-Atg7 (Sigma-Aldrich, A2856; IB 1:1000); goat polyclonal anti-cathepsin B [H-5] (Santa Cruz, sc-365558; IB 1:500; IF 1:100); rabbit polyclonal anti-p62 (Santa Cruz, sc-28359; IB 1:1000); rabbit polyclonal anti-Beclin-1 (Sigma-Aldrich, A00023; IB 1:1000); rabbit polyclonal anti-Cx43 (Invitrogen, 710700; IF 1:250); rabbit polyclonal anti-LC3 (Thermo Fisher Scientific, PA1-16930; IB 1:1000); Anti-Goat IgG (H + L), highly cross-adsorbed, CF™ 680 antibody produced in donkey (Sigma-Aldrich, SAB4600196; IF 1:500); Anti-EHD [E-8] (Santa Cruz, sc-390513; IB 1:1000); Anti-Eps15 [H-896] (Santa Cruz, sc-1840; IB 1:1000); TFEB (Cell Signalling, 4240, IF 1:100). HRP-conjugated goat anti-mouse IgG (Thermo Fisher Scientific, 31430; IB 1:10,000); HRP-conjugated mouse anti-mouse IgG (Biorad, 62650; IB 1:10,000); HRP-conjugated goat anti-rabbit IgG (Biorad; 656120; IB 1:10,000); Alexa488-conjugated donkey anti-goat IgG (Thermo Fisher Scientific, A-11055; IF 1:500); Alexa647-conjugated donkey anti-rabbit IgG (Thermo Fisher Scientific, A-31573; IF 1:500); Alexa568-conjugated donkey anti-mouse IgG (Thermo Fisher Scientific, A10037; IF 1:500); Alexa647-conjugated donkey anti-mouse IgG (Thermo Fisher Scientific, A-31571; IF 1:500).

## DNA constructs

Rat Cx43 cDNA was cloned into the KpnI and BamHI sites of a modified pENTR GFP C2 vector (Lopes et al, 2007). Site-directed mutagenesis was performed to generate mutants from Cx43 cDNA, followed by cloning into the KpnI and BamHI (Cx43$^{Y286A}$) or EcoRI and SalI (Cx43$^{Y265A}$, Cx43$^{Y265/286A}$) sites of the same vector. V5-Cx43 was generated by subcloning Cx43 into the KpnI and BamHI sites of a pENTR V5 C1 vector. Site-directed mutagenesis was performed to generate the Cx43$^{S368A}$ and Cx43$^{S368D}$ mutants from Cx43 cDNA, followed by cloning into the EcoRI and SalI sites of the same vector. V5-Cx43$^{Δ243}$ was generated by PCR amplification of Cx43 cDNA with the insertion of a stop codon after position 243, followed by subcloning into the BglII and KpnI sites of a pEGFP C2 vector. The GFP tag was then replaced with a V5 using NheI and BglII sites. GFP-Cx43 was generated by cloning Cx43 cDNA into the KpnI and BamHI sites of a pENTR GFP C1 vector. GFP-Cx43$^{Y265A}$ and GFP-Cx43$^{Y286A}$ were generated by cloning mutant Cx43 cDNA into the EcoRI and SalI sites of a pcDNA ENTR BP GFP C2 vector. All constructs were verified by DNA sequencing. Cx43$^{KKMP}$ was generated by sequential PCR by which the native stop codon of Cx43 was replaced with the sequence AAGAA-GATGCCTTAA, followed by cloning into the EcoRI and SalI sites of a pENTR vector. Cx43$^{Δ373}$ was generated by PCR amplification of Cx43 cDNA with the insertion of a stop codon after position 373, followed by cloning into the EcoRI and SalI sites of a pENTR vector. pmCherry-Gal3 was a gift from Hemmo Meyer (Addgene plasmid # 85662) (Papadopoulos et al, 2017). pcDNA3.2-GJA1-20K-V5 was a gift from Robin Shaw (Addgene plasmid # 49859) (Smyth and Shaw, 2013). pLifeAct_mScarlet_N1 was a gift from Dorus Gadella (Addgene plasmid # 85054) (Bindels et al, 2016). pLVX-PURO-GFP-ArpC2 was a gift from Edgar Gomes (Instituto de Medicina Molecular João Lobo Antunes, Faculdade de Medicina da Universidade de Lisboa).

## Cell culture

HEK293A parental cell line were purchased from the ATCC (American Type Culture Collection). Human embryonic kidney HEK293A cells were cultured in Dulbecco's modified Eagle's medium (DMEM, Thermo Fisher Scientific, Gibco, 12800082) supplemented with 10% FBS (Pan Biotech; P40-47500) and Penicillin/Streptomycin (100 U/ml:100 μg/ml), at 37 °C under 5% $CO_2$. HEK293$^{Cx43+}$ cell line was established as previously described (Catarino et al, 2011). Briefly, 1 day after HEK293A parental cells were infected with the lentiviral vector pLenti6-prom-CMV-V5-C1-Cx43, transduced cells were selected with 8 μg/ml blasticidin. A monoclonal cell line was then established by the limiting dilution method. For all the cell lines, mycoplasma contamination was assessed routinely to ensure that cells were mycoplasm-free.

Control and Niemann-Pick patient fibroblasts were obtained as previously described (Yambire et al, 2019). The breast cancer cell line SUM159PT, wild type and Cx43-knockout, were kindly provided by Trond Aasen (CIBER de Cáncer (CIBERONC), Instituto de Salud Carlos III, Madrid, Spain and Universitat Autònoma de Barcelona, Bellaterra, Spain) (Tishchenko et al, 2020). The HeLa cell line where all Atg8 family members were knocked-out (Atg8$^{KO}$) and the corresponding control cell line, were kindly provided by Michael Lazarou (Monash Biomedicine Discovery Institute, Monash University, Victoria, Australia) (Nguyen et al, 2016).

## Cell transfection and siRNA-mediated knockdown

DNA and siRNA transfections were performed using Lipofectamine 2000 according to the manufacturer's instructions. siRNA target sequences were Alix (Dharmacon, siGENOME SMARTpool M-004233-02-0005) and Arp2 (Thermo Fisher Scientific, s94586). 20 nmol/L siRNA was complexed with the transfection reagent and an overnight incubation at 37 °C was performed. After that, transfection media was replaced with fresh media and the experiments performed after 72 h (Alix) or 48 h (Arp2). Non-targeting control sequences (Ambion) were used as controls.

## Adenoviral vector production and cell transduction

HEK293A cells were incubated with the lentiviral vectors and 8 μg/mL of polybrene for 20 min at room temperature. After 30 min centrifugation at $800 \times g$ and 32 °C, cells were plated and monitored for the expression of GFP. Lentiviral vectors containing shRNA targeting ATG7 and the control empty vector were kindly provided by Dr. A.M. Cuervo (Albert Einstein College of Medicine, Yeshiva University, New York, USA).

## Quantification of LAMP1 at the plasma membrane by flow cytometry

Cells treated with LLOMe or with ionomycin and $Ca^{2+}$ were placed on ice, washed with flow cytometer buffer (PBS supplemented with 1% FBS and 2 mM EDTA), and then incubated for 30 min with anti-mouse Alexa Fluor-647 LAMP1 antibody. Before analysis, permeabilized cells were stained with 30 μg/mL PI. Data were acquired in a BD FACScantoTMII flow cytometer (Becton Dickinson): PI fluorescence was measured in the PI channel (585/42 BP) and LAMP1 in the Alexa647 channel (665/20 BP). The software for acquisition was BD FACSDiva, and data from at least 5000 cells were analysed with FlowJo. All PI-positive cells were excluded for the quantification. Representative images were obtained by primary antibody incubation for 20 min on ice. After wash, cells were fixed with PFA 4% and incubated with secondary antibody for 30 min and mounted in Mowiol for imaging.

## Biotinylation of cell-surface proteins

HEK293$^{Cx43+}$ cells grown on 35 mm culture dishes were rinsed twice with 2 mL ice-cold PBS containing 0.5 mM $MgCl_2$ and 1 mM $CaCl_2$, followed by the addition of 1 mL ice-cold solution containing 1 mg/mL (freshly added) Sulfo-NHS-SS-biotin (Pierce, Rockford, IL, USA). After 30 min at 4 °C, to stop subcellular trafficking, the medium was discarded and the plates were washed three times with PBS containing 0.5 mM $MgCl_2$, 1 mM $CaCl_2$ and 100 mM glycine. The cells were scraped in lysis buffer (50 mM Tris-HCl, 150 mM NaCl, 1% NP-40, 0.1% SDS, supplemented with protease inhibitor cocktail, 2 mM PMSF, 10 mM iodoacetamide, pH 7.5). After 15 min on ice, the cells were sonicated and the homogenates were centrifuged at 14,000 rpm, for 10 min. Supernatants were then transferred to 1.5 mL Eppendorf microfuge tubes containing 25 μL of Neutravidin beads (Pierce, Rockford, IL, USA). After 2 h incubation at 4 °C under agitation, the beads were washed four times with lysis buffer. The final pellets were resuspended in 20 μL 2x Laemmli buffer and denatured at 100 °C for 5 min. The beads were pelleted, and the solubilised proteins were analysed by Western blot.

## *Salmonella* infection and CFU assays

Cells were infected with 3–4 h (late logarithmic phase) subcultures of *Salmonella enterica* subsp. *enterica* (strain SL1344; DSMZ - German Collection of Microorganisms and Cell Cultures) at a multiplicity of infection (MOI) of 50. To synchronize infection, cells were centrifuged during 5 min at $500 \times g$, and then incubated for 25 min at 37 °C, 5% $CO_2$. Non-internalized extracellular bacteria were removed with two PBS washes and cells were further incubated with fresh culture medium without antibiotic during the experimental period. Bacteria cells viability was then evaluated using a colony forming unit (CFU) assay. The supernatants, corresponding to expelled/extracellular *Salmonella*, were collected and the adhered cells were then lysed with 0.5% Triton X-100 in sterile distilled water and scrapped (fraction corresponding to intracellular bacteria). Serial dilutions were performed, and each condition was spread in nutrient agar plates, with colony counting after 24 h at 37 °C.

## Transmission electronic microscopy

Extracellular vesicles were fixed with 2% PFA and allowed to absorb on Formvar-carbon coated niquel grids (TAAB Laboratories) for 5 min. Negative staining was then performed with 1% uranyl acetate, for 1 min. For immunogold staining, extracellular vesicles were adsorbed on Formvar-carbon-coated niquel grids for 30 min. Following glycine (50 mM) quenching, blocking buffer (1% BSA with 0.01% saponin) was added for 30 min. Extracellular vesicles were incubated with rabbit anti-Cx43 (1:50) and mouse anti-LAMP1 (1:50) overnight at 4 °C, washed and labelled with secondary antibodies conjugated to gold particles.

Isolated lysosomes were allowed to absorb on Formvar-carbon-coated copper grids (TAAB Laboratories) for 5 min.

HEK293A or HEK293[Cx43+] cells treated with LLOMe or CK869 were fixed with 2.5% glutaraldehyde in 0.1 M sodium cacodylate buffer (Agar Scientific, pH 7.2) supplemented with 1 mM calcium chloride. Sequential post-fixation was performed using 1% osmium tetroxide (Sigma), for 1.5 h, and contrast enhanced with 1% aqueous uranyl acetate, for 1 h. After rinsing in distilled water, samples were embedded in 2% molten agar and then dehydrated in a graded ethanol series (50–100%), impregnated and embedded using an Epoxy embedding kit (Fluka Analytical). Ultrathin sections (70 nm) were mounted on copper grids (300 mesh) and stained with lead citrate 0.2%, for 7 min.

All observations were carried out using a FEI-Tecnai G2 Spirit Bio Twin at 100 kV.

## Isolation of lysosomal enriched pool

The isolation of lysosomal enriched pool was performed according to the manufacturer's instructions from the Lysosomal isolation kit.

## Extracellular vesicles isolation

Cells were first cultured in extracellular vesicle-free medium for 24 h before being treated with LLOMe for 1 h. Media was then collected from cells and extracellular vesicles were isolated by ultracentrifugation. Media was subjected to differential centrifugation at 4 °C, starting with a 10 min centrifugation at $1000 \times g$ to remove cell debris, followed by 20 min at $16,500 \times g$ to isolate large extracellular vesicles (100 to 1000 nm in diameter, microvesicles). A final ultracentrifugation at $86,000 \times g$ for 70 min was carried out to isolate small extracellular vesicles (50 to 150 nm in diameter, exosomes). EVs were resuspended in RIPA buffer (150 mM NaCl, 50 mM Tris, 1% (w/v) NP-40 and 0.1% (w/v) SDS, pH 7.5), before samples were denatured and examined by Western blot.

## Immunofluorescence and cell imaging

After treatment, cell lines grown on gelatine-coated coverslips were fixed in 4% (w/v) PFA for 10 min, followed by 10 min permeabilization with PBS with 0.05% (w/v) saponin, and blocking with 1% (w/v) BSA for 20 min. Coverslips were then incubated overnight with primary antibodies diluted in the blocking solution at 4 °C. Next, cells were washed three times with PBS and incubated with Alexa Fluor-conjugated secondary antibodies for 1 h, at room temperature. Nuclei were stained with DAPI. Finally, coverslips were mounted with Mowiol 4-88. Antibody specificity was confirmed through immuno-blot analysis and secondary antibody controls, where primary antibodies were not added. Images were collected on a confocal microscope (Zeiss LSM 710; Carl Zeiss AG) by using a Plan-Apochromat 63x/1.4 oil DIC M27 objective. At least 5 different fields were imaged for each experimental condition and, for a given staining, the images were acquired with the same settings.

## Laurdan generalized polarization measurements

Cells were cultured on an Ibidi μ-slide 8 well at 60,000 cells/well. After 24 h, and before imaging, cells were incubated with 5 μM Laurdan for 10 min at 37 °C. In LLOMe-treated cells, Laurdan incubation was

carried out in the last 10 min. Membrane fluidity was then assessed by monitoring Laurdan fluorescence spectral shifts using two-photon excitation microscopy. Briefly, cells were imaged on a Leica TCS SP5 inverted microscope (DMI6000, Leica Microsystems CMS GmbH, Mannheim, Germany), with a 63× water apochromatic objective (1.2 NA). Two-photon excitation was performed with a Ti:sapphire laser (Mai Tai, Spectra-Physics, Darmstadt, Germany), setting the excitation wavelength to 780 nm. Laurdan fluorescence emission was collected at 400–460 nm ($I_{400-460}$) and 470–530 nm ($I_{470-530}$), and the respective generalized polarization (GP) value per pixel was determined as follows (Owen et al, 2012):

$$GP = \frac{I_{400-460} - GI_{470-530}}{I_{400-460} + GI_{470-530}}$$

The calibration factor G accounts for differences in the experimental setup and was obtained by measuring Laurdan fluorescence in DMSO ($GP_{DMSO} = 0.0357$) using the same acquisition conditions as those set for living cells. Background contribution was subtracted from both channels. All image analysis was implemented in MATLAB (Mathworks, Natick). In each cell, only regions of interest (ROI) at the plasma membrane were selected for mean GP determination. For each condition, at least 400 cells were analysed, measured during 3 independent experiments.

## AFM imaging

AFM studies were conducted using a NanoWizard IV atomic force microscope (JPK Instruments, Berlin, Germany). The AFM is mounted on a Zeiss Axiovert 200 inverted optical microscope. HEK293A and HEK293[Cx43+] cells were cultured on Petri dishes (35 mm). LLOMe treatment was performed 60 min before starting the AFM experiments.

All cell conditions were fixed with a 2% (v/v) glutaraldehyde solution. Imaging of the cells was performed in air, in contact mode. Oxidized sharpened silicon tips (ACL, Applied NanoStructures, Inc., Mountain View, CA, USA) with a spring constant of 3 N/m and a resonance frequency of 60 kHz were used for the imaging. Scanning speed was adjusted to 0.35 Hz and acquisition points were 512 × 512 pixels. Imaging data were analysed with the JPK Image Processing software v. 6.055 (JPK Instruments, Berlin, Germany) (Carvalho et al, 2010; Ribeiro et al, 2016).

## Cell elasticity

Differences in HEK293A and HEK293[Cx43+] cells elasticity (with and without LLOMe treatment) were evaluated by AFM-based force spectroscopy measurements. This methodology was previously described by us (Chaffin et al, 1998) (Guedes et al, 2019, 2016; Bernardes et al, 2016). For these experiments, the softest triangular cantilevers of OMCL TR-400-type silicon nitride AFM probes (Olympus, Tokyo, Japan) were used, with a resonance frequency of approximately 3 kHz in solution and a tip radius of 15 nm. The spring constants of the cantilevers (values close to 0.02 N/m) were calibrated by the thermal fluctuation method. The maximum applied force was adjusted to 500 pN before retraction. For each condition tested, approximately 440 cells were tested, and 5 force vs. distance curves were collected on each cell. By applying the

Hertzian model, the curves were fitted and the Young's modulus data were obtained, using JPK Image Processing software v. 6.055. Values of AFM tip indentation depth into the cells for an applied force of 500pN were also obtained.

## Image analysis

Image analysis was performed on the original data using Fiji Image J1 software (version 1.53q). Colocalization was quantified using ImageJ plugin Colo2. Pearson's correlation coefficient was assessed either per transfected cell with tagged proteins (GFP-Cx43, GFP-Arp2 and mCherry-Gal3) or by immunostaining (LAMP1). The background was subtracted from the total fluorescence intensity of each image. For actin network assessment, z-stack images were acquired, and the network was extracted by using LPX Filter2d plugin (Higaki, 2017; Biel et al, 2020). The filter "lineFilters" and the linemode "lineExtract," were used to skeletonize each selected cell. The settings for "lineExtract" used were giwslter = 1, mdnmsLen = 8, shaveLen = 5, delLen = 5, preGauss = −1. Next, Skeleton-Analyse Skeleton (2D/3D) tools were used to analyse the branches.

Total mCherry-Gal3 and GFP-Arp2 puncta were measured by applying a threshold to each cell, and the regions of interest (ROIs) were detected by the Analyse Particles plugin from Fiji ImageJ. Puncta in clusters were separated using the watershed function. In mCherry-Gal3 experiments, a normalization relative to the initial value of total puncta intensity was performed: The total intensity of mCherry puncta from each time point after washout was divided by the Gal3 puncta after 1 h of LLOMe.

Lysosome positioning within cells was measured as previously described (Domingues et al, 2023). Briefly, cell membrane and nuclear contours were extracted manually using ImageJ selection tools. Lysosome location and coordinates were also detected by drawing the ROI of each lysosome. The XY coordinates obtained were exported for posterior analysis using a custom MATLAB script. The measurement axis for each lysosome was defined as a straight line crossing the nucleus and the lysosome centroids, originating the final coordinates for the calculation. All calculations and the appropriate ratios were obtained using Matlab. Data was binned and plotted by fraction of lysosome per distance to plasma membrane.

## Cathepsin B release

After treatments, cell supernatants were collected and centrifuged to remove cellular contamination. Cath B was measured using the Cath B ELISA kit according to the manufacturer's indication.

## Immunoprecipitation

Cell lysates were prepared in RIPA buffer, supplemented with protease inhibitors, 2 mM phenylmethanesulfonyl fluoride (PMSF), 10 mM iodoacetamide and 2 mM sodium orthovanadate. Immunoprecipitations were performed using 300 μg of total protein lysates, with an overnight incubation with primary antibodies (0.5 μg goat anti-Cx43; 0.5 μg rabbit anti-Arp2), at 4 °C under rotation. Protein G Sepharose beads were then added and incubated for 1.5 h, at 4 °C. After washing, complexes were eluted in Laemmli buffer and analysed by Western blot.

## Western blot

Cell extracts were prepared with Laemmli buffer by sonication and boiling 5 min at 95 °C. Samples were separated through sodium dodecyl sulfate polyacrylamide gel electrophoresis (SDS-PAGE) and transferred to nitrocellulose membranes. Ponceau staining was performed for total protein evaluation. After ponceau wash, membranes were blocked with 5% (w/v) non-fat milk in Tris-buffered saline-Tween 20 (TBST; 20 mM Tris, 150 mM NaCl, 0.2% (v/v) Tween 20, pH 7.6). Primary antibodies were incubated overnight at 4 °C, followed by HRP-conjugated secondary antibodies for 1 h, at room temperature. Each protein was detected by chemiluminescence using an ImageQuant LAS 500 from GE Healthcare Life Sciences. Quantifications were performed in unsaturated images using Image Lab software (Bio-Rad).

## Lysosome pH measurement

Lysosome pH was measured after an overnight incubation with dextran conjugated with FITC (250 μg/mL) and dextran conjugated with Alexa Fluor 647 (50 μg/mL) followed by a 3 h chase before treatment with 0.5 mM of LLOMe or with 100 nM bafilomycin for 1 h. Cells were live-imaged by confocal microscopy. To measure the fluorescence intensity ratio between FITC- and Alexa Fluor 647-dextrans, a threshold was applied to define the lysosomes through the analyse particle macro from ImageJ Fiji software. Using the obtained ROIs, the signal intensity of each individual lysosome was measured in each channel. After background removal the fluorescence intensity ratio of the two channels was calculated.

## Autophagic flux

Autophagic flux was evaluated using data acquired from the immunoblot analysis of LC3B in cells in the presence and absence of bafilomycin A1 (Baf). HEK293A and HEK293[Cx43+] were treated with 0.5 mM of LLOMe for 60 min and then incubated for 120 min with 100 nM Baf. Autophagic flux was calculated by dividing the ratio of LC3B-II/LC3B-I in cells treated with Baf by the same ratio from untreated cells.

## DQ-BSA degradation

Cells were loaded with 50 μg/mL DQ-BSA for 4 h. After the pulse, HEK293A and HEK293[Cx43+] were washed out and treated with LLOMe. The DQ-BSA degradation was assessed by measuring the fluorescence of the DQ-BSA fragments using a BioTek Synergy HT multidetection microplate reader (BioTek Instruments), excitation 530/25 nm and emission 590/35 nm. Fluorescence was recorded during 180 min.

## Live-cell imaging

For live imaging, cells were seeded in Lab-Teck plates from Nunc. After transfection or cellular treatments, cells were then incubated with complete DMEM without phenol red supplemented with 10 mM HEPES. Live cells were imaged at 37 °C using the LSM710 confocal system and a 63x/1.4NA objective. Z-stack images were acquired 60 min after LLOMe treatment. For pH measurements, a

single stack was acquired and analysed. The movies were performed by time-lapse imaging, through the acquisition of one z-stack image per 1 min for a total of 10 to 15 min.

## siRNA mediated knockdown

siRNA targeting Eps15 (s4773 (CAGCAUUCUUGUAAACGGA) and s4774 (CUACCUUACUAGCCCAUAU)) and a non-targeting control sequence were obtained from Ambion (Silencer Select Pre-designed siRNA). siRNA targeting Erp29 (sc-60599) was obtained from Santa Cruz Biotechnology. siRNA targeting Alix (siGENOME SMARTpool M-004233-02-0005) was obtained from Dharmacon. siRNA targeting Arp2 (s94586) was obtained from Thermo Fisher Scientific. Cells were grown until they reach 40–50% confluency. siRNA was complexed with Lipofectamine 2000 (Invitrogen), according to manufacturer's recommendations, and added to the cell media at a final concentration of 20 nM for each Eps15 siRNA and 40 nM for all other siRNA and control sequence. Experiments were performed 48 h after transfection.

## Beads transfection

Beads (Polysciences, 17010-5) preparation and transfection were performed as previously reported (Kobayashi et al, 2010). Briefly, transfection reagent-coated beads were prepared by mixing the beads with Effectene Transfection Reagent (Qiagen) according to the manufacturer's instructions, with bead suspension replacing DNA solution. HEK293A cells were incubated with bead complexes for one hour. After that, cells were washed twice with PBS and incubated with fresh medium for an additional 3 h before fixation.

## Lucifer Yellow

Dye uptake, as a measure of hemichannel activity, was tested using Lucifer Yellow (LY). For that, HEK293A cells transfected with wild type (Cx43$^{WT}$) or Cx43 mutants (Cx43$^{S368A}$ and Cx43$^{S368D}$) were incubated with 1 mg/mL LY and 1 mg/mL rhodamine dextran (RD; MW 10 kDa) diluted in Ca$^{2+}$ and Mg$^{2+}$-free uptake solution (Ca$^{2+}$/Mg$^{2+}$-free HBSS [5.3 mM KCl, 0.4 mM KH$_2$PO$_4$, 4.2 mM NaHCO$_3$, 138 mM NaCl, 0.3 mM Na$_2$HPO$_4$ and 5.6 mM D-Glucose at pH 7.4] supplemented with 10 mM EGTA) for 10 min at 37 °C, followed by washing with PBS/Ca$^{2+}$/Mg$^{2+}$ and fixation in 4% PFA for 10 min at room temperature. Images were acquired, and LY fluorescence per cell was quantified using Fiji software, excluding cells positive for RD. Data are presented as the percentage of positive cells, with threshold defined by the median LY fluorescence intensity of cells expressing Cx43$^{WT}$.

## Statistics analysis

With the exception of the supportive experiments Appendix Figs. S2e, S3h,i, S3l, S4j and S5m which were performed twice, each experiment was repeated at least three times with similar results, using independent experimental samples and statistical tests as specified in the figure legends. Replicates were obtained from different passages of the same cell line. The statistical analysis to estimate significance was performed following the recommendation of Graphpad Prism version 8.0.1.

## Data availability

This study includes no data deposited in external repositories.

The source data of this paper are collected in the following database record: biostudies:S-SCDT-10_1038-S44318-024-00177-3.

## Peer review information

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

## Acknowledgements

This work was supported by the European Regional Development Fund (ERDF) through the Operational Program for Competitiveness Factors (COMPETE) under the projects: PPBI-POCI-01-0145-FEDER-022122, UIDB/04539/2020, UIDP/04539/2020, project FCT-PTDC/MED-NEU/8030/2020, FCT-2022.09311, 'la Caixa' Foundation/HR22-00854, John Black Charitable Foundation and Wellcome Trust Award in Science/224361/Z/21/Z, Horizon 2020 grant 857524, and the Comissão de Coordenação e Desenvolvimento Regional do Centro - CCDRC through the Centro2020 Programme. This work was partially supported by the H2020 Twinning project RESETageing (GA 952266).

## Author contributions

**Neuza Domingues**: Conceptualization; Data curation; Formal analysis; Investigation; Methodology; Writing—original draft. **Steve Catarino**: Conceptualization; Validation; Investigation; Writing—review and editing. **Beatriz Cristóvão**: Investigation. **Lisa Rodrigues**: Methodology. **Filomena A Carvalho**: Methodology. **Maria João Sarmento**: Methodology. **Mónica Zuzarte**: Methodology. **Jani Almeida**: Methodology. **Teresa Ribeiro-Rodrigues**: Validation; Investigation; Methodology. **Ânia Correia-Rodrigues**: Validation; Methodology. **Fábio Fernandes**: Resources. **Paulo Rodrigues-Santos**: Resources. **Trond Aasen**: Resources. **Nuno C Santos**: Data curation; Formal analysis; Investigation; Methodology. **Viktor I Korolchuk**: Conceptualization; Formal analysis; Writing—review and editing. **Teresa Gonçalves**: Formal analysis; Investigation; Methodology; Writing—review and editing. **Ira Milosevic**: Conceptualization; Resources; Formal analysis; Writing—review and editing. **Nuno Raimundo**: Conceptualization; Resources; Formal analysis; Writing—review and editing. **Henrique Girão**: Conceptualization; Data curation; Formal analysis; Supervision; Funding acquisition; Validation; Project administration; Writing—review and editing.

Source data underlying figure panels in this paper may have individual authorship assigned. Where available, figure panel/source data authorship is listed in the following database record: biostudies:S-SCDT-10_1038-S44318-024-00177-3.

## Disclosure and competing interests statement

The authors declare no competing interests.

