## [Peer Review File · The EMBO Journal]

Connexin43 promotes exocytosis of damaged lysosomes through actin remodelling

Henrique Girao, Neuza Domingues, Steve Catarino, Beatriz Cristovao, Lisa Rodrigues, Filomena Carvalho, Maria Joao Sarmento, Monica Zuzarte, Jani Almeida, Teresa Ribeiro-Rodrigues, Ania Correia-Rodrigues, Fabio Fernandes, Paulo Rodrigues-Santos, Trond Aasen, Nuno Santos, Viktor Korolchuk, Teresa Gonçalves, Ira Milosevic, and Nuno Raimundo

Corresponding author(s): Henrique Girao (hmgirao@fmed.uc.pt)

Review Timeline:

Submission Date:	11th Dec 23
Editorial Decision:	2nd Feb 24
Revision Received:	30th Apr 24
Editorial Decision:	29th Jun 24
Revision Received:	4th Jul 24
Accepted:	9th Jul 24

Editor: William Teale

Transaction Report:

Dear Henrique,

Thank you again for sharing your work and the submission of your manuscript (EMBOJ-2024-116370) to The EMBO Journal. Please accept my sincere apologies for the unusually long peer-review period take for your study. Your manuscript was sent to three reviewers for evaluation: we have received reports from two of them, which I enclose below. Please note that while feedback from referee #3 is still pending at this stage we have, in light of the other reviewers' input, decided to proceed with our decision in order to expedite the manuscript's processing.

As you will see, both referees acknowledge the potential interest and novelty of your work, as well as the level of robustness and clarity needed for publication in The EMBO Journal.

I judge the comments of the referees to be generally reasonable and given their overall interest, we are in principle happy to invite you to revise your manuscript to address the referees' comments. This decision must be, however, contingent on there being no technically overriding concerns presented by referee #3 (whose comments I will share as soon as I receive them).

I should add that it is The EMBO Journal policy to allow only a single major round of revision and that it is therefore important to resolve the main concerns at this stage. I believe the concerns of the referees are reasonable and addressable, but please contact me if you have any questions, need further input on the referee comments or if you anticipate any problems in addressing any of their points. Please, follow the instructions below when preparing your manuscript for resubmission.

I would also like to point out that as a matter of policy, competing manuscripts published during this period will not be taken into consideration in our assessment of the novelty presented by your study ("scooping" protection). We have extended this 'scooping protection policy' beyond the usual 3 month revision timeline to cover the period required for a full revision to address the essential experimental issues. Please contact me if you see a paper with related content published elsewhere to discuss the appropriate course of action.

Again, please contact me at any time during revision if you need any help or have further questions.

Thank you very much again for the opportunity to consider your work for publication. I look forward to your revision.

Best regards,

William

William Teale, Ph.D.
Editor
The EMBO Journal

When submitting your revised manuscript, please carefully review the instructions below and include the following items:

- 1) a .docx formatted version of the manuscript text (including legends for main figures, EV figures and tables). Please make sure that the changes are highlighted to be clearly visible.
- 2) individual production quality figure files as .eps, .tif, .jpg (one file per figure).
- 3) a .docx formatted letter INCLUDING the reviewers' reports and your detailed point-by-point response to their comments. As part of the EMBO Press transparent editorial process, the point-by-point response is part of the Review Process File (RPF), which will be published alongside your paper.
- 4) a complete author checklist, which you can download from our author guidelines ([https://wol-prod-cdn.literatumonline.com/pb-assets/embo-site/Author Checklist%20-%20EMBO%20J-1561436015657.xlsx](https://wol-prod-cdn.literatumonline.com/pb-assets/embo-site/Author%20Checklist%20-%20EMBO%20J-1561436015657.xlsx)). Please insert information in the checklist that is also reflected in the manuscript. The completed author checklist will also be part of the RPF.
- 5) Please note that all corresponding authors are required to supply an ORCID ID for their name upon submission of a revised manuscript.

6) We require a 'Data Availability' section after the Materials and Methods. Before submitting your revision, primary datasets produced in this study need to be deposited in an appropriate public database, and the accession numbers and database listed under 'Data Availability'. Please remember to provide a reviewer password if the datasets are not yet public (see <https://www.embopress.org/page/journal/14602075/authorguide#datadeposition>). If no data deposition in external databases is needed for this paper, please then state in this section: This study includes no data deposited in external repositories. Note that the Data Availability Section is restricted to new primary data that are part of this study.

Note - All links should resolve to a page where the data can be accessed.

8) For data quantification: please specify the name of the statistical test used to generate error bars and P values, the number (n) of independent experiments (specify technical or biological replicates) underlying each data point and the test used to calculate p-values in each figure legend. The figure legends should contain a basic description of n, P and the test applied. Graphs must include a description of the bars and the error bars (s.d., s.e.m.).

9) We would also encourage you to include the source data for figure panels that show essential data. Numerical data can be provided as individual .xls or .csv files (including a tab describing the data). For 'blots' or microscopy, uncropped images should be submitted (using a zip archive or a single pdf per main figure if multiple images need to be supplied for one panel). Additional information on source data and instruction on how to label the files are available at .

10) We replaced Supplementary Information with Expanded View (EV) Figures and Tables that are collapsible/expandable online (see examples in <https://www.embopress.org/doi/10.15252/embj.201695874>). A maximum of 5 EV Figures can be typeset. EV Figures should be cited as 'Figure EV1, Figure EV2" etc. in the text and their respective legends should be included in the main text after the legends of regular figures.

12) Our journal encourages inclusion of *data citations in the reference list* to directly cite datasets that were re-used and obtained from public databases. Data citations in the article text are distinct from normal bibliographical citations and should directly link to the database records from which the data can be accessed. In the main text, data citations are formatted as follows: "Data ref: Smith et al, 2001" or "Data ref: NCBI Sequence Read Archive PRJNA342805, 2017". In the Reference list, data citations must be labeled with "[DATASET]". A data reference must provide the database name, accession number/identifiers and a resolvable link to the landing page from which the data can be accessed at the end of the reference. Further instructions are available at .

Further instructions for preparing your revised manuscript:

We realize that it is difficult to revise to a specific deadline. In the interest of protecting the conceptual advance provided by the work, we recommend a revision within 3 months (2nd May 2024). Please discuss the revision progress ahead of this time with the editor if you require more time to complete the revisions. Use the link below to submit your revision:

Referee #1:

The study by Domingues et al. reported a new mechanistic role of Cx43 in mediating the exocytosis of damaged lysosome through its association with actin associated proteins. They demonstrated that Cx43 was recruited to damaged lysosome, leading to an increase in their secretion. This process is mediated by the interaction of Cx43 with the actin nucleator protein Arp2, resulting in actin rearrangement and lysosomal exocytosis. The presented data are comprehensive and thorough, addressing several key mechanisms. However, the explanation of how Cx43 mediates this process in a channel-independent manner is less convincing. Additionally, western blots need to be quantified, and the quality of many images needs improvement with a better solution and magnification.

Specific comments.

1. Western blots need to be repeated and quantified, including Fig. 1e, Fig. 3f, Fig. 6h. The quality of western blots in Fig. 3f (LAMP1) is poor and needs improvement.
2. In the main text, stably expressing Cx43 in HEK293 cells was used, which is an overexpression system and protein trafficking and assembly could vary in this system. Although in supplemental data, a cancer cell line endogenously expressing Cx43 was adopted, it would be useful to focus on endogenous Cx43 and present those data in the main figures.
3. In Fig. 2, the expulsion of Salmonella appears to be much greater than LAMP1. Does this possibly suggest that lysosome may not be a major mechanism for Salmonella excretion?
4. In Fig. 4a, no explanation was provided regarding why CPZ doesn't work, while other clathrin-mediated and caveolae-dependent endocytosis inhibitors work.
5. Fig. 4b needs quantification. Also it's better to show single-labeled images with colors.
6. Mutants Cx43S368A and S368D are used to explore channel-dependent or independent function. However, there is no data showing if these mutants actually work on gap junctions/hemichannels in the cells as expected. Additionally, there is no study in the paper to show the effect of gap26 on channels.
7. It's very difficult to see the changes in Fig. 5c comparing 293A with 293 with Cx43.
8. Images are blurry in Fig. 6a, d, e, and g, and the quantitative data generated are questionable.
9. Fig. 6h, why LLOMe treatment does not seem to alter GFP-Arp2, which is unexpected?

Referee #2:

The manuscript reports that lysosome damage triggers lysosomal exocytosis via an autophagy-independent mechanism (Fig. 1). From here, the authors decided, using a rationale that does not appear to be very stringent, to investigate the role of connexin 43 (Cx43) in this process. The reason justifying this choice is that Cx43 interacts with lysosome-related machineries (ESCRT) and can reach the lysosomes, mainly, as far as was known up to this, for degradation. Instead, the authors hypothesize that Cx43 plays an active role at lysosomes and that it is not transported there just for degradation.

To test their hypothesis, they perform a complete and complex series of experiments on a cell line overexpressing Cx43 (HEK293-Cx43). In fact, all the experiments reported in the manuscript (with the exception of few data obtained on SUM159PT cells, Fig. S2c, d), were performed in HEK293-Cx43. These cells present a clear gain of function in terms of lysosomal exocytosis upon lysosome damage indicating that over-expressed Cx43 may in fact induce lysosomal exocytosis upon LLOME treatment. By investigating the mechanism of action by which Cx43 controls lysosomal exocytosis, the authors find that it does not involve its "channel" activity but its ability to locally remodel the actin cytoskeleton via interaction with ARP2.

The major concern of this referee is to what extent this model could be relevant for the actual role of endogenous Cx43 in lysosomal exocytosis induced by LLOME and hence also the relevance of the results on the mechanism of action of Cx43 in lysosomal exocytosis performed under conditions of artificially-induced high levels of Cx43. In fact, no experiment was designed to assess the role of the endogenous Cx43 on lysosomal exocytosis or actin remodelling in HEK293 cells, a cell line that as the authors state "present negligible levels of endogenous Cx43". Surprisingly, what is missing is a quantitative comparison of the levels of Cx43 in the different cell models: the levels of the endogenous Cx43 in HEK293 compared to those in Cx43 overexpressing cells or SUM159PT cells.

Another concern regards the lysosome targeting and topology of Cx43 under LLOME. It is known, also thanks to previous work by the authors, that Cx43 is targeted to lysosomes for degradation. This targeting involves ubiquitination of Cx43 and incorporation of Cx43 into the inner vesicles of MVBs. Indeed, Cx43 can interact with the ESCRT machinery which plays a pivotal role in cargo selection and in the inward budding of vesicles inside MVBs. This process allows the degradation of Cx43 once MVBs fuse with lysosomes, but also the release of Cx43-containing extracellular vesicles (EVs) when MVBs fuse with the PM. The authors report that Cx43 is enriched in EVs released from LLOME-treated cells and in MVBs of these cells, and this would imply that LLOME would enhance the "normal" trafficking of Cx43 to MVBs. However, if this is the case then Cx43 should reside in the inner vesicles of MVBs and then in the lumen of lysosomes. From this location, Cx43 would be unable to interact with cytosolic components such as ARP2/actin.

How do the authors reconcile this topological inconsistency?

Specific comments:

Fig. 1. The quantification in c does not match the profile in b since 10 min LLOME here is less than ionomycin;
e: the basal signal for surface galectin 3 and LAMP1 is rather high and not consistent with surface LAMP1 in 1b or 1c.
b: Yellow arrows(?) are indicated in the legend but are missing.

Fig. 2

g: large dispersion of data with just two points different in HEK293 Cx-43 as compared to wt HEK293

Fig. 3

f: the amount of LAMP1 that co-IPs at 60 min is not different from the amount detected in the blank (no Ab)

Fig4

b: where is Cx43 relocated in BFA-treated cells? There is no signal at all for Cx43.

c: surprisingly, there are many Gal3 puncta. One would expect these to be much less considering the graph in Fig. 2a where LLOME induces only a tiny (less than 20%) increase in Gal3-positive structures or the image in Fig. S2b where very rare Gal3 puncta are seen. In addition, the expression of the mutant Delta243 appears to have fewer Gal3 puncta, contrasting with the hypothesis that a Cx43 mutant would be unable to induce exocytosis.

Fig. S1 Panel f: the level of surface LAMP1 should be normalized for the levels of a protein resident at the PM.

Fig. S3 Panel c: the HEK293-Cx43 cells have a lower rate of maturation of CatB, which becomes insensitive to BAF.

Fig. S4: Although the authors claim that there are no differences in the autophagy pathway in HEK293-Cx43 cells, clear differences do exist judging from Fig. S4f as differences in p62 under LLOME or BAF (decreased compared to HEK293) and Beclin under the same treatments (increased compared to HEK293).

Fig. S5 Panel I: experiments assessing the half-life of Cx43 using cycloheximide. The concentration reported in the legend is 5 mM CHX. I am sure this is a mistake. If the levels of Cx43 are normalized for calnexin levels (used as normalizer), then LLOME appears to significantly impair the degradation of Cx43 (even more than BAF).

Referee #1 (Report for Author)

The study by Domingues et al. reported a new mechanistic role of Cx43 in mediating the exocytosis of damaged lysosome through its association with actin associated proteins. They demonstrated that Cx43 was recruited to damaged lysosome, leading to an increase in their secretion. This process is mediated by the interaction of Cx43 with the actin nucleator protein Arp2, resulting in actin rearrangement and lysosomal exocytosis. The presented data are comprehensive and thorough, addressing several key mechanisms.

However, the explanation of how Cx43 mediates this process in a channel-independent manner is less convincing.

We fully acknowledge the Reviewer's comment and to improve the quality and robustness of our data we have performed additional experiments. In the original version of the manuscript, we tested if alterations in Cx43 channel activity impacted the exocytosis capacity of cells after lysosomal damage induction by measuring cell surface LAMP1 (Fig. 4 of the main manuscript). We used two different strategies: pharmacological (GAP26) and the expression of two different Cx43 mutants that form constitutively open (Cx43 S368A) or closed channels (Cx43 S368D). Our data shows that neither blocking nor opening Cx43 channels significantly affected the levels of lysosomal exocytosis upon LLOMe treatment. This suggests that Cx43 channel activity is not required for lysosomal exocytosis following damage. The strategies and tools used in this work were validated and used in previous studies of our group¹. However, to confirm these results in the present experimental setup, we performed additional experiments to assess channel activity reduction by either GAP26 or the mutation of serine residue 368 (S368D) of Cx43, by measuring the entrance of the Lucifer yellow (LY) dye in cells. Our new data, included in Fig. S7a and b and displayed here in **Figure 1a**, reveal that HEK293A cells expressing Cx43 S368D present around 50% less LY staining, when comparing to cells expressing the wildtype form of Cx43, confirming that this mutation is effective in reducing Cx43 channel activity. On the other hand, as expected, when the same Ser is replaced by an Ala the channels are more active. Moreover, we also confirmed the efficacy of GAP26 as a Cx43 channel inhibitor, in the HEK293^{Cx43+} cell line. Indeed, we observed a reduction of about 50% in LY staining after treatment with GAP26, corroborating previous data showing that this mimetic peptide inhibits Cx43 channel activity.

Figure 1 – Hemichannel function was assessed by permeation to Lucifer Yellow (LY) and cells were imaged by widefield microscopy. Fluorescence per cell was quantified and data is presented by percentage of positive cells with threshold defined by Cx43^{WT} median fluorescence. $n = 5$ (minimum 200 cells per condition).

We have now added the following sentence to the revised version of the manuscript:

“To address this question, cells were transfected with Cx43 mutants that form constitutively open (Cx43^{S368A}) or closed channels (Cx43^{S368D}), after which channel activity was assessed using lucifer yellow assay (Fig. S7a).”

We also included additional details about the methodology employed for this experiment.

Additionally, western blots need to be quantified, and the quality of many images needs improvement with a better solution and magnification.

We thank the Reviewer for the observation, and we do apologize for not having been clear. In the submitted version we performed the quantification of all the main results obtained from western blots analysis as indicated: Fig. 1e – quantification in Fig. S1 c and d; Fig.2 k – quantification in Fig. 2 l and m; Fig 3 f – quantification in Fig. 3 g; Fig. 6 h – quantification in Fig. S10e. Additionally, all western blots in the supplementary data whose information was required for data interpretation were quantified, as in Fig. S3 a-d, Fig. S4, Fig. 5 l – quantification in Fig. 5 m.

We had not quantified data from western blots used to validate some experimental conditions, as in the silencing of specific proteins, overexpression or drug treatment, which are already well established in the literature. Nevertheless, we have chosen to include the blots to trustworthy support other results that depend on the validation of these approaches.

Almost all microscopy images have an associated inset with a higher magnification, with the exception of images already presented with higher zoom. We also use arrows to highlight points of interest. Nevertheless, to address the issues raised by the Reviewer, in the revised version of the manuscript higher magnifications were added to Fig. 5c and Fig. 6a.

Specific comments.

1. Western blots need to be repeated and quantified, including Fig. 1e, Fig. 3f, Fig. 6h. The quantification of Figure 1e is displayed in Fig S1c and d (see the following cropped figure)

The results of Figure 3f are also quantified in Fig. 3g.

LAMP1 was used as a positive control of a “general” lysosomal protein to confirm that we were isolating Cx43 localized at the lysosome. Considering that Cx43 and LAMP1 interaction was not the focus of our work, we did not quantify this co-immunoprecipitation. Furthermore, the interaction of Cx43 with LAMPs, in the context of different forms of autophagy, namely CMA, is an ongoing project in the lab.

Concerning the WB of Figure 6h, we also performed this quantification, and the results were plotted in Figure S10e.

Fig. 6h.

The quality of western blots in Fig. 3f (LAMP1) is poor and needs improvement.

We acknowledge the comment of the Reviewer, however, although the LAMP1 immunoblot appears to be poor, due to their high glycosylation levels, LAMP protein immunoblots often present a smeared band, as published in several others scientific studies:

Ratto, E. et al. Direct control of lysosomal catabolic activity by mTORC1 through regulation of V-ATPase assembly. doi:10.1038/s41467-022-32515-6.

Singh, J. et al. Cross-linking of the endolysosomal system reveals potential flotillin structures and cargo. doi:10.1038/s41467-022-33951-0.

Eriksson, I., Wäster, P. & Öllinger, K. Restoration of lysosomal function after damage is accompanied by recycling of lysosomal membrane proteins. <https://doi.org/10.1038/s41419-020-2527-8>.

Zhao, S. et al. Salmonella effector SopB reorganizes cytoskeletal vimentin to maintain replication vacuoles for efficient infection. doi:10.1038/s41467-023-36123-w.

2. In the main text, stably expressing Cx43 in HEK293 cells was used, which is an overexpression system and protein trafficking and assembly could vary in this system. Although in supplemental data, a cancer cell line endogenously expressing Cx43 was adopted, it would be useful to focus on endogenous Cx43 and present those data in the main figures.

We thank the Reviewer for this valuable and pertinent comment. We do apologise if the lack of clarity in presenting our data led to a misunderstanding. In the submitted version of the manuscript, besides SUM159PT cancer cell line, we had also used human fibroblasts with endogenous expression of Cx43 (FigureS5 g-k). Indeed, to support the idea that the effect of Cx43 on lysosome exocytosis is not an artifact of Cx43 overexpression, we had performed some key experiments in fibroblasts from patients with Niemann-pick type A (NPA1) and C (NPC1), with endogenous levels of Cx43, to reinforce the importance of Cx43 during the cellular response to lysosomal damage. In addition to endogenously expressing Cx43, Niemann-Pick diseases are part of a group of autosomal recessive lysosomal storage disorders (LSD), which are characterized by the lysosomal accumulation of different lipid species, leading to their loss of function and membrane integrity²⁻⁴. As previously shown, in basal conditions NP-patients' fibroblasts present higher levels of Gal3 colocalization with LAMP1, being exacerbated by additional treatment with LLOMe (**Fig S5g-i**). Interestingly, higher levels of Cx43 and Gal3/LAMP1

colocalization in patients' fibroblasts were also observed, with a further augment induced by LLOMe treatment. In parallel, we measured lysosomal exocytosis in this cellular model by assessing cell surface LAMP1. We found higher levels of basal exocytosis in NPC1 fibroblasts, as previous suggested^{2,5} (**Fig. S5j-k**). Moreover, upon LLOMe treatment, all fibroblast cell lines displayed increased levels of LAMP1 at cell surface, with the levels in NPC patient's fibroblast being significantly higher than control cells. Noteworthy, to further strengthen the role of Cx43 in lysosomal exocytosis in conditions of loss of lysosomal membrane integrity, we have now included LAMP1 levels at cell surface in human fibroblasts depleted of Cx43 (**Figure 2 from this letter, and new Fig. S5l**). The new data shows that silencing Cx43 in fibroblasts from patients with NPA1 or NPC1, induces a significant decrease in cell surface LAMP1 levels at basal conditions. This result corroborates the importance of Cx43 in lysosomal exocytosis in response to lysosomal damage. Furthermore, we also observed that human fibroblast from patients present increased levels of Cx43 when compared with control cells.

We have now also included the following sentence in the manuscript:

“Interestingly, these findings were also followed by an increase in exocytosis (**Fig. S5j-k**). In contrast, Cx43 depletion in NPA1 and NPC1 fibroblasts resulted in a reduction of LAMP1 at cell surface (**Fig. S5l**). Of note, NPA1 and NPC1 fibroblasts present in basal conditions increased levels of Cx43 (**Fig. S5m**)”

Figure 2 – Quantification of LAMP1 levels at cell surface measure by flow cytometry in human fibroblasts from a control healthy donor, patients with Niemann-pick type A (NPA1) and C (NPC1), transfected with siCt or siRNA against Cx43 (left). On the right panel, immunoblot of Cx43 protein levels in human fibroblasts.

To reinforce the robustness of the data obtained in HEK293Cx43 cells, we also analyzed the actin cytoskeleton network in NPA1 and NPC1. Our data clearly shows a different actin organization in NPA1 and NPC1 (**Figure 3**), with increased branched and patched actin structures and increased Arp2 puncta distribution. Interestingly, we observed lysosomes colocalizing with patches of actin and Cx43 near the cell border as indicated by the yellow arrows (**Figure 3a**). Moreover, NPA1 and NPC1 present increased levels of Cx43 near Arp2 puncta structures (**Figure 3b**).

Figure 3 – Actin network in human fibroblasts. (a and b) Confocal images of control human fibroblasts and fibroblasts from patients with NPA1 and NPC1 immunostained for Cx43 and LAMP1 (a), or Cx43 and Arp2 (b). Actin was stained by phalloidin. In (a), arrows highlight the colocalization areas of Cx43, Lamp1 and actin patches structures. In (b), yellow arrows point actin patches and blue arrows highlight the colocalization areas of Cx43, Arp2 and actin.

We have added these new data in the supplementary data (Fig S9c-d). Additionally, the following sentences were included in the manuscript:

“Importantly, this increase in Arp2 recruitment to actin filaments was also found around engulfed *Salmonella* in infected-HEK293^{Cx43+} (Fig 6d), as well as in lysosomes from NPA1 and NPC1 fibroblasts (Fig. S9c). NPA1 and NPC1, which presented a branched and patched actin cytoskeleton, also demonstrated Cx43 recruitment to actin positive lysosomes near the plasma membrane (Fig. S9d).”

Importantly, using the RAW 264.7 macrophage cell line, which endogenously expresses Cx43, we have recently studied the effect of Cx43 recruitment to macrophage phagosomes containing *Candida albicans*⁶. This pathogen induces loss of phagosomal membrane impermeability upon its transition from yeast to hyphae. We demonstrated that Cx43 is recruited to expanding phagosomes and promotes hyphae folding, a recently described host defence mechanism dependent on actin polymerization⁷.

Although in supplemental data, a cancer cell line endogenously expressing Cx43 was adopted, it would be useful to focus on endogenous Cx43 and present those data in the main figures.

Although we do understand the Reviewer's suggestion to move the data obtained from the cancer cell line endogenously expressing Cx43 to the main text, instead of it being a part of the supplementary data, we would prefer to maintain the order of the Figures as presented in the initial version of the manuscript to facilitate data interpretation. Considering the large amount of data that we have produced to properly tackle the questions raised by the Reviewers, we believe that including all the validation experiments, obtained from the cancer cell line and human fibroblasts to the main figures would increase the complexity of our Figures and obscure the main scientific message that we want to transmit.

3. In Fig. 2, the expulsion of Salmonella appears to be much greater than LAMP1. Does this possibly suggest that lysosome may not be a major mechanism for Salmonella excretion?

We acknowledge the pertinence of this issue and thank the Reviewer for having addressed it. In our opinion, a direct comparison between the value ranges from CFU measurements and cell surface LAMP1 cannot be performed. To be accurate, we can only state that both techniques show increased secretion. However, a plausible explanation for the differences can reside in the fact that Salmonella can divide inside lysosomes, as it can be appreciated in the figures, and the fusion of one phagolysosome with the plasma membrane can release several pathogens into the extracellular medium, resulting in several colony forming bacteria.

4. In Fig. 4a, no explanation was provided regarding why CPZ doesn't work, while other clathrin-mediated and caveolae-dependent endocytosis inhibitors work.

We thank the Reviewer for having pinpointed this pertinent question. Although we don't have a perfect explanation for this observation, we can provide some speculation based on reports by other authors. Although widely used as an endocytic inhibitor, the effect of CPZ on endocytosis was reported to vary among different cell lines, and in some cases it can lead to increased internalization of membrane proteins⁹. A study using Gn-11, a cell line derived from mouse LHRH neurons and astrocytes, showed that after 1h of treatment with 10 μ M of CPZ (the same concentration used in our work), there was a reduction in cell-cell contacts and a redistribution of Cx43, resulting in a diffuse intracellular localization that was very intense in vesicle-like structures¹⁰. Despite the well established general role of CPZ in endocytosis inhibition, this work suggests, that depending on cell type, dose and incubation time, CPZ may elicit other non-specific cellular responses, affecting protein distribution. Furthermore, it was also demonstrated that CPZ not only impairs the autophagy-lysosome pathway¹¹ but can also induce lysosomal membrane permeabilization¹². Considering these results, CPZ can be

promoting an increase in LAMP1 and Cx43 colocalization through a mechanism independent of its effect on clathrin-mediated endocytosis.

5. Fig. 4b needs quantification.

We apologize if we were not clear enough in the manuscript text. The quantifications for Fig. 4b from the main figures were presented in Fig. S6 g-h.

Also it's better to show single-labeled images with colors.

We apologise if the interpretation of the results seems less obvious if the individual channels are presented in black and white. In our opinion the best way to display individual channels is in black and white since it provides the best contrast for the human eye. In order to better show our results, we would like to keep the individual panels in black and white as follow by other studies published in EMBO Journal:

Baba, T. et al. 2019; Phosphatidylinositol 4,5-bisphosphate controls Rab7 and PLEKHM1 membrane cycling during autophagosome–lysosome fusion; <https://doi.org/10.15252/emj.2018100312>

6. Mutants Cx43S368A and S368D are used to explore channel-dependent or independent function. However, there is no data showing if these mutants actually work on gap junctions/hemichannels in the cells as expected. Additionally, there is no study in the paper to show the effect of gap26 on channels.

We thank the Reviewer for having raised this relevant issue. As we mention in the beginning of this letter, although these approaches have been already used in previous studies, including in our group, we agree that it is important to confirm them in our

experimental setup. To tackle this question, we overexpressed the two mutated forms of Cx43 or treated cells with GAP26, after which we assessed channel activity through the entrance in cells of the dye Lucifer yellow (LY). Our data confirms that both treatment with GAP26 or mutation of serine residue 368 to aspartic acid (S368D) inhibit Cx43 channel activity. Indeed, we show that HEK293A cells expressing Cx43S368D present around 50% less LY staining, when compared to cells expressing the wildtype form of Cx43, confirming that this mutation is effective in reducing Cx43 channel activity. On the other hand, as expected, when the same serine residue is replaced by an alanine the channels are more active. Moreover, we also investigated the efficacy of GAP26 as a Cx43 channel inhibitor, in the HEK293Cx43 cell line, where we observed a reduction of about 50% in LY staining after treatment with GAP26, corroborating previous data showing that this mimetic peptide inhibits Cx43 channel activity. These results were included in Fig. S7a and b of the revised version of the manuscript and displayed here in **Figure 1a**.

Figure 2 – Hemichannel function was assessed by permeation to Lucifer Yellow (LY) and cells were imaged by widefield microscopy. Fluorescence per cell was quantified and data is presented by percentage of positive cells with threshold defined by Cx43^{WT} median fluorescence. n = 5 (minimum 200 cells per condition).

7. It's very difficult to see the changes in Fig. 5c comparing 293A with 293 with Cx43.

We acknowledge this relevant issue identified by the Reviewer. The differences are not immediately obvious and require an experienced and trained eye. However, we do believe that these differences can be perceived, and we have pinpointed them with arrows. Furthermore, we have applied a plugin that allowed us to quantify these changes in an unbiased manner. In addition, to make them easier to discern, in the revised version of the manuscript we have included magnified insets in each panel.

8. Images are blurry in Fig. 6a, d, e, and g, and the quantitative data generated are questionable.

We appreciate this comment, but it is important to mention that Arp2 presents a cytosolic distribution and only displays a dotted pattern when it is recruited to initiate actin

nucleation. Thus, in control conditions, Arp2 is dispersed in the cytosol and the images seem blurrier, as demonstrated in other publications:

Dong, Y et al.; De novo actin polymerization is required for model Hirano body formation in Dictyostelium; doi:10.1242/bio.014944

LeClaire, L. et al.; Phosphorylation of the Arp2/3 complex is necessary to nucleate actin filaments; doi: 10.1083/jcb.200802145

Of note, the Arp2 images were acquired in the same conditions as all others fluorescence images.

It is important to emphasize that quantification of these images was performed in an unbiased manner resorting to an Fiji software plugin. The background was defined, and the threshold performed to identify and define the Arp2-dotted pattern, which was then quantified using the particle analysis plug in.

9. Fig. 6h, why LLOMe treatment does not seem to alter GFP-Arp2, which is unexpected?

We acknowledge the importance of this apparent inconsistency and thank the Reviewer for having highlighted it. The data presented in the manuscript clearly show that Arp2 interacts with Cx43, namely in the presence of cytochalasin, where this interaction is more obvious. However, intriguingly, LLOMe-treated cells do not present increased levels of interaction between Arp2 and Cx43. Indeed, based on the increased levels of Cx43-Arp2 colocalization quantified by immunofluorescence imaging in cells treated with LLOMe, we would expect increased levels of interaction also by co-immunoprecipitation (co-IP) of Cx43 and Arp2.

We have repeated this experiment with slight technical changes in order to try to preserve weak protein-protein interactions. The results obtained are highly consistent, and a clear increased interaction between Cx43 and Arp2 was observed when cells are treated with cytochalasin D. In spite of this, we believe that the data obtained from colocalization between Cx43 and Arp and co-IP of these two proteins are not incompatible. Given the limited resolution of fluorescence microscopy, colocalization studies demonstrate that the two proteins co-exist in the same area, being interacting or not, whereas co-IP means that the Cx43 and Arp2 are part of the same protein complex. It is also likely that, for some reason, the interaction in basal conditions is weaker and more labile than in the cells treated with cytochalasin, being lost during co-IP assays. To strengthen this hypothesis, and reinforce co-IP data, we assessed the colocalization of Cx43 and Arp2, by confocal microscopy, upon cytochalasin D treatment. The results presented in **Figure 5** of this rebuttal show that the dotted pattern of Arp2 is more pronounced and there is a higher colocalization of Arp2 with Cx43 in cytochalasin-treated cells. (see **Figure 5** for the Reviewers), which is in agreement with our data of increased co-immunoprecipitation. Thus, overall, our findings show for the first time that after lysosomal damage or when actin-nucleation is inhibited (cytochalasin D), Cx43 is recruited to Arp2-mediating actin nucleation patches, where the two proteins likely interact. Strikingly, these results are in agreement with a recent pre-print showing a new role for the actin nucleation machinery, namely Arp2/3 in maintaining lysosomal integrity (Theodore, C. et al; Autophagosome turnover requires Arp2/3 complex-mediated

Figure 5- Fluorescence confocal images of Arp2 and Cx43 in HEK293Cx43+ transfected with GFP-Arp2. Cells were control and LLOMe-treated cells were treated with cytochalasin D (Cyto D). Yellow arrows point for colocalization point of Cx43 and Arp2.

Referee #2 (Report for Author)

The manuscript reports that lysosome damage triggers lysosomal exocytosis via an autophagy-independent mechanism (Fig. 1). From here, the authors decided, using a rationale that does not appear to be very stringent, to investigate the role of connexin 43 (Cx43) in this process. The reason justifying this choice is that Cx43 interacts with lysosome-related machineries (ESCRT) and can reach the lysosomes, mainly, as far as was known up to this, for degradation. Instead, the authors hypothesize that Cx43 plays an active role at lysosomes and that it is not transported there just for degradation. To test their hypothesis, they perform a complete and complex series of experiments on a cell line overexpressing Cx43 (HEK293-Cx43). In fact, all the experiments reported in the manuscript (with the exception of few data obtained on SUM159PT cells, Fig. S2c, d), were performed in HEK293-Cx43. These cells present a clear gain of function in terms of lysosomal exocytosis upon lysosome damage indicating that over-expressed Cx43 may in fact induce lysosomal exocytosis upon LLOMe treatment. By investigating the mechanism of action by which Cx43 controls lysosomal exocytosis, the authors find that it does not involve its "channel" activity but its ability to locally remodel the actin cytoskeleton via interaction with ARP2.

The major concern of this referee is to what extent this model could be relevant for the actual role of endogenous Cx43 in lysosomal exocytosis induced by LLOME and hence also the relevance of the results on the mechanism of action of Cx43 in lysosomal exocytosis performed under conditions of artificially-induced high levels of Cx43. In fact, no experiment was designed to assess the role of the endogenous Cx43 on lysosomal exocytosis or actin remodelling in HEK293 cells, a cell line that as the authors state "present negligible levels of endogenous Cx43".

We thank the Reviewer for the valuable comments and pertinent criticism.

Protein overexpression is an important and widely used tool to bring new mechanistic insights about the biological functions of specific proteins. Nevertheless, we agree that only using artificially-induced high levels of Cx43 in our experiments by using the overexpression of Cx43 on HEK293A would not be enough to clearly and definitively demonstrate the role of Cx43 on lysosomal exocytosis upon damage. We now reproduced key experiments in other cell lines to strengthen our hypothesis and validate our model. In addition to HEK293 cells, as referred by the Reviewer, we also used the breast cancer cell line SUM159PT (Figure S2a-c and S7e), both a control cell line, with endogenous Cx43 expression and a Cx43-depleted cell line. Additionally, **in the submitted version of the manuscript, we included a cellular model of fibroblasts from patients with Niemann-pick type A (NPA1) and C (NPC1)** (Fig. S5g-k from the submitted version). In addition to endogenously expressing Cx43, Niemann-Pick diseases are part of a group of autosomal recessive lysosomal storage disorders (LSD), which are characterized by the lysosomal accumulation of different lipid species, leading to their loss of function and membrane integrity²⁻⁴. As previously shown, in basal conditions NP-patients' fibroblasts present higher levels of Gal3 colocalization with LAMP1, which is exacerbated by treatment with LLOMe (**Fig. S5g-i** from the supplementary data). Interestingly, higher levels of Cx43 and Gal3/LAMP1 colocalization in patients' fibroblast were also observed, with a further increase in cells subjected to LLOMe treatment. In parallel, we measured lysosomal exocytosis in this cellular model by assessing LAMP1 at the cell surface. We found higher levels of basal exocytosis in NPC1 fibroblasts, as previously suggested^{2,5} (**Fig. S5j-k** from the supplementary data). Moreover, upon LLOMe treatment, all fibroblast cell lines increased their levels of LAMP1 at cell surface, with the levels in NPC1 patient's fibroblasts being significantly higher than control cells.

In order to comprehensively address the concerns of the Reviewer we performed important additional experiments in these fibroblast cell lines. Thus, in the revised version of the manuscript, we quantified the levels of LAMP1 at cell surface in fibroblasts silenced for Cx43. Noteworthy, the results depicted here (**Figure 6a**) and in Fig. S5l from the supplementary data demonstrate that depletion of Cx43 from human fibroblast diminished the levels of LAMP1 from NPA1 and NPC1 cells, becoming not significantly different from the control. We also investigated actin cytoskeleton alterations in NPA1 and NPC1. Our new data (**Figure 6b** here and Fig. S10 in the supplementary data) shows that control fibroblasts present long stress fibers, whereas NPA1 and NPC1 cells present several branches and patches actin structures, similar to what we observe in HEK293Cx43 cells after LLOMe treatment. Importantly, we observed an accumulation of Cx43 in lysosomes localized near the plasma membrane where actin patches structures were also detected (**Figure 6c** here and Fig. S10 in the supplementary data). Moreover,

NPA1 and NPC1 fibroblasts also present increased levels of Cx43 near Arp2-positive actin patches (Figure 6c). Of note, endogenous Cx43 levels in patient fibroblasts (which present lysosomal damage) are higher than in control cells (Figure 6d).

Figure 6- (a) Quantification of LAMP1 levels at cell surface measure by flow cytometry in human fibroblasts from a control healthy donor, and patients with Niemann-pick type A (NPA1) and C (NPC1), transfected with a non-specific control siRNA or siRNA against Cx43. (b-c) Actin network in human fibroblasts. (a and b) Confocal images of control human fibroblasts and fibroblast from patients with NPA1 and NPC1 immunostained for Cx43 and LAMP1 (a), or Cx43 and Arp2 (b). Actin was stained by phalloidin. Yellow arrows from (a) point LAMP1, Cx43 and actin colocalization areas and in (b) highlight the actin patches structures. In b, blue arrows point colocalization areas of Arp2, actin and Cx43. (d) Immunoblot of Cx43 protein levels in the human fibroblasts.

We have now included the following sentence in the revised version:

“Importantly, this increase in Arp2 recruitment to actin filaments was also found around engulfed *Salmonella* in infected-HEK293^{Cx43+} (Fig 6d), as well as in lysosomes from

NPA1 and NPC1 fibroblasts (Fig. S9c). NPA1 and NPC1, which presented a branched and patched actin cytoskeleton, also demonstrated Cx43 recruitment to actin positive lysosomes near the plasma membrane (Fig. S9d).”

Surprisingly, what is missing is a quantitative comparison of the levels of Cx43 in the different cell models: the levels of the endogenous Cx43 in HEK293 compared to those in Cx43 overexpressing cells or SUM159PT cells.

We thank the Reviewer for the valuable comment. We now include in the revised version of the manuscript representative immunoblots of Cx43 levels in the different cell lines, including HEK293A (Fig. S2a) and SUM159PT (Fig. S2b), Figure 7.

Figure 7 - Immunoblot images of Cx43 levels of HEK293A and HEK293^{Cx43+}, SUM159PT WT and Cx43-KO cells. Calnexin and Ponceau were used as a loading control.

Another concern regards the lysosome targeting and topology of Cx43 under LLOME. It is known, also thanks to previous work by the authors, that Cx43 is targeted to lysosomes for degradation. This targeting involves ubiquitination of Cx43 and incorporation of Cx43 into the inner vesicles of MVBs. Indeed, Cx43 can interact with the ESCRT machinery which plays a pivotal role in cargo selection and in the inward budding of vesicles inside MVBs. This process allows the degradation of Cx43 once MVBs fuse with lysosomes, but also the release of Cx43-containing extracellular vesicles (EVs) when MVBs fuse with the PM. The authors report that Cx43 is enriched in EVs released from LLOME-treated cells and in MVBs of these cells, and this would imply that LLOME would enhance the "normal" trafficking of Cx43 to MVBs. However, if this is the case then Cx43 should reside in the inner vesicles of MVBs and then in the lumen of lysosomes. From this location, Cx43 would be unable to interact with cytosolic components such as ARP2/actin.

How do the authors reconcile this topological inconsistency?

We thank the Reviewer for having raised this highly relevant question, that had also intrigued us.

To better clarify our model, we performed a schematic representation of Cx43 cellular trafficking (Figure 8 in this letter), showing the Cx43 endocytic distribution and its N- and C-terminal orientation along the pathway.

Figure 8 – Schematic representation of Cx43 trafficking in the endolysosomal pathway.

Based on literature (reviewed in ^{13–15} and the results published by our group ¹⁶, plasma membrane localized Cx43 presents both the N- and C-terminal turned to cytoplasmic side, which is required to recruit the endocytic machinery that will drive Cx43 internalization. Later, during endocytic vesicle maturation, Cx43 will end up in MVB, where Cx43 localized to i) intra-luminal vesicles (ILV), with both termini facing the ILV lumen, or ii) MVB limiting membrane, with the N- and C-terminal oriented towards the cytoplasm. If the MVB fuses with the plasma membrane, MVB content is released into the extracellular milieu, giving rise to EV of endocytic origin (commonly known as exosomes). However, if the MVB instead fuses with the lysosome, Cx43-containing ILV will be degraded inside the lysosome, whereas Cx43 localized on the MVB membrane is maintained on the lysosome membrane, with both N- and C-termini turned to cytosol. This structural orientation enables Cx43 to interact with its cytosolic interactors/partners. Accordingly, our confocal and TEM images show that Cx43 localizes to the lysosomal membrane. Our results using several mutants of Cx43, such as Cx43 mutated in AP2 binding sites and a C-terminal truncated isoform (which show a reduction of Cx43 colocalization with damaged lysosomes, with impact in lysosomal exocytosis (**Fig. 4**) and cytoskeleton organization (**Fig. S7k**)), support this model. Additionally, our data using the Cx43 mutant truncated in the last 9 amino acids, which has been described as an actin binding site ¹⁷, presented in **Fig. S7l-o** of the submitted revised version, corroborate the importance of the Cx43 C-terminal in the cellular response to lysosomal damage. Altogether, we propose that the C-terminal of Cx43 present on the lysosomal membrane (and turned towards the cytosol) anchors the actin machinery necessary for cortical actin remodelling, which facilitates the access of lysosomes to the PM and consequently, increases their exocytosis.

Specific comments:

Fig. 1. The quantification in c does not match the profile in b since 10 min LLOME here is less than ionomycin;

	Sample Name	Subset Name	Mean : Comp-APC-A
■	lamp_A Ct.fcs	Live cells	40.3
■	lamp_ION+ca2+.fcs	Live cells	95.5
■	lamp_LLOMe 10min.fcs	Live cells	109
■	lamp_LLOMe 60min.fcs	Live cells	141
■	lamp_LLOMe 180min.fcs	Live cells	120

Figure 9 - Representation of Figure 1b in 2D. Arrows indicate the deviation of curves, showing an increase in counts with higher LAMP1 levels at cell surface. Table was elaborated directly from FLOWJO software, showing the mean for each histogram.

We do apologise if the way we presented the data caused a misunderstanding. To clarify this question, we show the same histogram of Fig. 1b from the main figures of the submitted manuscript without perspective (**Figure 9**). The blue curve from LAMP1 levels after HEK293A incubation with ionomycin and calcium and the different LLOMe conditions curves present a similar width. However, LLOMe-treated cells show higher counts with more LAMP1 intensity. The arrows highlight the curves deviation for higher counts in LLOMe-treated cells with higher levels of LAMP1 intensity.

e: the basal signal for surface galectin 3 and LAMP1 is rather high and not consistent with surface LAMP1 in 1b or 1c.

We apologize for any confusion regarding the Reviewer's concern and we thank the opportunity to try to clarify it. To evaluate cell surface levels of LAMP1, we employed two different methods—flow cytometry, using a specific antibody targeting LAMP1 proteins, and biotinylation of plasma membrane proteins, followed by pull down and immunoblot analysis. Despite yielding slightly different results, both techniques reveal similar variations in LAMP1 levels at the cell surface after LLOME treatment, with a fivefold increase observed in flow cytometry (Fig. 1c) and a threefold increase in biotinylation assays (Supplementary Fig. S1c). However, differences between the two experimental approaches can be explained by technical issues and limitations of each methodology. In the flow cytometry, since PI-positive cells are excluded from the quantification, the contribution of lysosomal LAMP1 staining due to the presence of permeable holes at the plasma membrane is eliminated. Moreover, in this technique, we are also measuring single cell fluorescence, thus we can speculate that this technique is more accurate and sensitive than the biotinylation assay. Although biotinylation can be considered a less sensitive technique, we believe that it would be important to present both assays to strengthen our data.

The presence of both LAMP1 and Galectin-3 at the plasma membrane in basal conditions was previously reported. Besides its role in lysosome repair/ degradation Galectin-3 is also a cell surface protein, with an important role in numerous processes, such as tumour cell adhesion and progression, inflammation and immunity, and wound healing^{18,19}. Originally, Galectin-3 was identified as Mac-2, a cell surface antigen presented on activated peritoneal macrophages²⁰. Thus, due to their conserved carbohydrate recognition domain, it is expected that depending on carbohydrate levels at cell surface, Galectin-3 will be bound to the plasma membrane. Furthermore, Galectin-3 has been implicated in a specific type of endocytosis, called Clathrin-independent Carrier (CLIC), where Galectin-3 functions as an endocytic adaptor that co-clusters glycosylated cargos and glycosphingolipids at the plasma membrane²¹. Additionally, several reports have shown that Galectin-3 can interact with lipids at the plasma membrane. On the other hand, the presence of LAMP1 at cell surface is not surprisingly since it has been reported in different cell models. For example, part of newly synthesized LAMP1 arrival to the lysosomes involves transport via the plasma membrane²².

b: Yellow arrows(?) are indicated in the legend but are missing.

We would like to thank the observation and apologise for the mistake. The sentence was written for Fig. 2b and we have now removed it from the revised version of the manuscript.

Fig. 2
g: large dispersion of data with just two points different in HEK293 Cx-43 as compared to wt HEK293

We understand the concerns of the Reviewer, but this technique presents a huge variability among experiments which justifies the dispersion. Nevertheless, it is noteworthy to mention that in each independent experiment, the HEK293Cx43 cell line

presented increased levels of extracellular *Salmonella*, with statistical difference when comparing with control cells.

Fig. 3
f: the amount of LAMP1 that co-IPs at 60 min is not different from the amount detected in the blank (no Ab)

We acknowledge the Reviewer for having raised this issue. The focus of this particular experiment was Gal3 and not LAMP1, aiming to show that upon lysosomal damage Cx43 is recruited to Gal3-positive injured lysosomes. As part of our focus on the interplay between Cx43 and lysosomal proteins, we have another ongoing study centred specifically on the crosstalk between Cx43 and LAMP1 and LAMP2a, showing that Cx43 is a substrate of chaperone-mediated autophagy (CMA) (data not shown, but we can provide some data if required). Consistently, we have always observed a decrease in the amount of LAMP1 co-IP with Cx43 after 60 min of treatment with LLOME. These results strongly suggest that upon lysosomal damage the interaction of LAMP1 with Cx43 increases in initial periods, after which it is likely disrupted. Since the highest levels of LAMP1 at the cell surface are detected 60 min after LLOME treatment, it is plausible that increased interaction of Cx43 with LAMP1 in early stages reflect the arrival of Cx43 to damaged lysosomes. Later, it is likely this interaction with LAMP1 in injured lysosomes is replaced by other partners, namely actin remodelling proteins required for exocytosis to the plasma membrane.

Fig4
b: where is Cx43 relocated in BFA-treated cells? There is no signal at all for Cx43.
In cells treated with BFA, Cx43 is mainly localized at the ER. Below the Reviewer can find the original uncropped images, where the Cx43 distribution is evident.

Figure 10 – Full cells from the images depicted in Figure 4b from the submitted version of the manuscript.

c: surprisingly, there are many Gal3 puncta. One would expect these to be much less considering the graph in Fig. 2a where LLOME induces only a tiny (less than 20%)

increase in Gal3-positive structures or the image in Fig. S2b where very rare Gal3 puncta are seen.

We thank the Reviewer for addressing this issue. However, first we would like to highlight that Fig.2a represents the levels of Gal3 after LLOMe washout. Additionally, Fig. S2b display the representative images of HEK293A and HEK293Cx43 after the washout. Importantly, the basal levels of Gal3 between both cell lines after 1h of LLOMe treatment are similar (Fig.S2a and the first panel of Fig.S2b). Nevertheless, we agree that the quantity of lysosomes positive to Gal3 in the Cx43-WT expressing cell in basal conditions is overrepresented in Figure 4c and we have now replaced it with a more representative one.

In addition, the expression of the mutant Delta243 appears to have fewer Gal3 puncta, contrasting with the hypothesis that a Cx43 mutant would be unable to induce exocytosis.

We agree and thank the Reviewer for having identified this apparent inconsistency. However, we did not notice significant differences between the Gal3 puncta distribution in the cells expressing the different Cx43 mutants. To address the Reviewer's concern, we quantified the total number of Gal3 positive lysosomes and total Gal3 intensity per cell in cells transfected with control Cx43, Cx43 Δ 243, Cx43^{Y286A} and Cx43^{Y265A}. **Figure 11** shows no statistical difference in these parameters between the different Cx43 mutants.

Figure 11 - Quantification of the total number of Galectin-3 (Gal3) positive lysosomes and the total intensity of Gal3 per cell. The results are mean \pm SEM of two independent experiments.

Fig. S1 Panel f: the level of surface LAMP1 should be normalized for the levels of a protein resident at the PM.

We apologize if we are not understanding the question raised by the Reviewer. As shown below, panel f from Fig. S1 does not present LAMP1.

Fig. S3 Panel c: the HEK293-Cx43 cells have a lower rate of maturation of CatB, which becomes insensitive to BAF.

We thank the Reviewer for raising this relevant question. In both cell lines we observed an increase in pro-CathB after 2h of Baf treatment, which can be explained by the loss of lysosomal acidic pH, since pro-CathB is cleaved to its mature form by acidic lysosomal proteases. Thus, alkalinization of the lysosomal lumen is expected to inhibit CathB maturation. Moreover, considering that CathB was shown to be required for Cx43 lysosomal degradation²³, the observed increase of pro-CathB is plausible to be related with the cellular response to increased Cx43 levels to efficiently promote Cx43 turnover. Importantly, lysosomal pH and degradative capacity was not affected by increased Cx43 levels (Fig S3 g-h). Furthermore, lysosomal number and size also seems to not be affected by increased Cx43 levels (**Figure 12** and now Fig. S3d-c).

Figure 12 - Quantification of lysosomal total number (left) and size (right) in HEK293A and HEK293Cx43+ cells in control or after 1h incubation with LLOMe. The results are the mean±SEM from three independent experiments. *** $p < 0.001$.

Fig. S4: Although the authors claim that there are no differences in the autophagy pathway in HEK293-Cx43 cells, clear differences do exist judging from Fig. S4f as differences in p62 under LLOMe or Baf (decreased compared to HEK293) and Beclin under the same treatments (increased compared to HEK293).

We acknowledge the relevance of this issue and thank the Reviewer for pointing it out. Although the panels we chose to construct Fig. S4f do show a decrease in p62 levels in HEK293Cx43 cells under LLOMe or Baf, when the data from several experiments were analyzed we did not observe any statistical difference among the different experimental conditions. To address the concerns of the Reviewer we carried out additional experiments to assess p62 and beclin1 levels in response to LLOMe and Baf treatment (Figure 13). Taking into account these new results for the statistical analysis, the lack of significant statistical differences is maintained. We removed the Galectin-3 western blot of this image considering that it decontextualized in the scope of these experiments to assess the impact of LLOMe on autophagic pathway. The main change of Galectin-3 after lysosomal damage/lysophagy, is its redistribution, displaying punctate structures instead of its usual cytosolic localization.

Having into consideration Reviewer concerns, we have replaced previous western blot by a new more representative one.

Figure 13 – Immunoblot image of p62 and beclin-1 in cells HEK293A and HEK293Cx43+ control, treated with LLOMe or bafilomycin A for 1 h.

To further reinforce our conclusions, we have also performed microscopy assays to evaluate the levels and subcellular localization p62 and TFEB, a master regulator of lysosomal biogenesis after its trafficking to the nucleus²⁴. As expected, conditions of lysosomal damage induced by LLOMe treatment potentiates TFEB recruitment to the nuclear region, however this new data (Fig. S3k-l, and here Figure 14) demonstrate no difference between the HEK293A and HEK293Cx43 cell lines in basal conditions and after LLOMe treatment.

Figure 14 - TFEB distribution in HEK293A or HEK293Cx43+ control cells or in cells incubated with LLOMe. On the right, quantification of mean intensity of nuclear TFEB. At least 300 cells were quantified in each condition. Scale bar, 10 μ m.

Regarding p62 immunostaining, in HEK293A and HEK293Cx43, our results, now presented in Fig. S4j-k and here in Figure 15, show no statistical difference between the total intensity of p62 puncta in both cell lines, after LLOMe treatment.

Figure 15 - Representative confocal images of p62 protein in control or LLOMe-treated HEK293A or HEK293Cx43+ cells. Graph represents the quantification of total puncta intensity per cell. At least 20 cells per condition were analysed.

Fig. S5 Panel I: experiments assessing the half-life of Cx43 using cycloheximide. The concentration reported in the legend is **5 mM CHX**. I am sure this is a mistake. If the levels of Cx43 are normalized for calnexin levels used as normalizer), then LLOMe appears to significantly impair the degradation of Cx43 (even more than BAF).

We thank the Reviewer for having identified this issue. Indeed, it was a mistake. We used 50µg/mL and the correspondent concentration in molar is 178 µM. We have now corrected the concentration in the revised version of the manuscript. Additionally, we also agree that the total calnexin is not the same in all conditions. However, the Cx43 levels decay in control and LLOMe-treated cells are much higher than the difference in calnexin protein (see the cropped images in **Figure 16a**). In BafA1 the Cx43 is more stable during the cycloheximide treatment, as our quantification have shown. Nevertheless, we are including a different representative image of the quantification (**Figure 16b**).

Figure 16 - Immunoblot image of Cx43 from HEK293^{Cx43+} treated with 178 μ M cycloheximide (CHX) for 1, 2 or 4 h, with or without LLOMe or Bafilomycin A1 (Baf). Calnexin was used as loading control. **a** Fig. 5m with different image rearrangement and **b** a different representative image of Cx43 levels in CHX experiment.

References

1. Martins-Marques, T. *et al.* EHD1 Modulates Cx43 Gap Junction Remodeling Associated with Cardiac Diseases. *Circ Res* **126**, E97–E113 (2020).
2. Kaufmann, A. M. & Krise, J. P. Niemann-Pick C1 functions in regulating lysosomal amine content. *Journal of Biological Chemistry* **283**, 24584–24593 (2008).
3. Davis, O. B. *et al.* NPC1-mTORC1 signaling Couples Cholesterol Sensing to Organelle Homeostasis and is a Targetable Pathway in Niemann-Pick type C HHS Public Access. *Dev Cell* **56**, 260–276 (2021).
4. Pascua-Maestro, R. *et al.* Apolipoprotein D-mediated preservation of lysosomal function promotes cell survival and delays motor impairment in Niemann-Pick type A disease. (2020) doi:10.1016/j.nbd.2020.105046.
5. Buratta, S. *et al.* Molecular Sciences Lysosomal Exocytosis, Exosome Release and Secretory Autophagy: The Autophagic-and Endo-Lysosomal Systems Go Extracellular. *Int. J. Mol. Sci* **2020**, 2576 (2576).
6. Beatriz Cristovao, Neuza Domingues & Henrique Girao. Cx43-mediated hyphal folding counteracts phagosome integrity loss during fungal infection. *Microbiol Spectr* (2023).

7. Bain, J. M. *et al.* Immune cells fold and damage fungal hyphae. *Proc Natl Acad Sci U S A* **118**, 1–8 (2021).
8. Victoria, G. S. & Zurzolo, C. The spread of prion-like proteins by lysosomes and tunneling nanotubes: Implications for neurodegenerative diseases. *Journal of Cell Biology* Preprint at <https://doi.org/10.1083/jcb.201701047> (2017).
9. Vercauteren, D. *et al.* The use of inhibitors to study endocytic pathways of gene carriers: Optimization and pitfalls. *Molecular Therapy* **18**, (2010).
10. Orellana, J. A., Palacios-Prado, N. & Sáez, J. C. Chlorpromazine reduces the intercellular communication via gap junctions in mammalian cells. *Toxicol Appl Pharmacol* **213**, (2006).
11. Li, N. *et al.* Chlorpromazine affects autophagy in association with altered Rag GTPase–mTORC1–TFEB signaling. *Front Cell Dev Biol* **11**, (2023).
12. Park, S. H., Hyun, J. Y. & Shin, I. A lysosomal chloride ion-selective fluorescent probe for biological applications. *Chem Sci* **10**, (2019).
13. Epifantseva, I., Shaw, R. M. & Claude Herve, J. Intracellular trafficking pathways of Cx43 gap junction channels ☆. (2017) doi:10.1016/j.bbamem.2017.05.018.
14. Leithe, E., Mesnil, M. & Aasen, T. The connexin 43 C-terminus: A tail of many tales ☆. (2017) doi:10.1016/j.bbamem.2017.05.008.
15. Martins-Marques, T. *et al.* Biological Functions of Connexin43 Beyond Intercellular Communication. *Trends in Cell Biology* vol. 29 835–847 Preprint at <https://doi.org/10.1016/j.tcb.2019.07.001> (2019).
16. Soares, A. R. *et al.* Gap junctional protein Cx43 is involved in the communication between extracellular vesicles and mammalian cells. *Sci Rep* (2015) doi:10.1038/srep13243.
17. Baum, R. *et al.* GJA1-20k, an internally translated isoform of Connexin 43, is an actin capping protein. doi:10.1101/2022.01.05.475034.
18. Argüeso, P. *et al.* Association of cell surface mucins with galectin-3 contributes to the ocular surface epithelial barrier. *Journal of Biological Chemistry* **284**, (2009).
19. Mazurek, N. *et al.* Cell-surface galectin-3 confers resistance to TRAIL by impeding trafficking of death receptors in metastatic colon adenocarcinoma cells. *Cell Death Differ* **19**, (2012).
20. Ho, M. K. & Springer, T. A. Mac-2, a novel 32,000 Mr mouse macrophage subpopulation-specific antigen defined by monoclonal antibodies. *The Journal of Immunology* **128**, (1982).
21. Lakshminarayan, R. *et al.* Galectin-3 drives glycosphingolipid-dependent biogenesis of clathrin-independent carriers. *Nat Cell Biol* **16**, (2014).
22. Rohrer, J., Schweizer, A., Russell, D. & Kornfeld, S. The targeting of lamp1 to lysosomes is dependent on the spacing of its cytoplasmic tail tyrosine sorting motif relative to the membrane. *Journal of Cell Biology* **132**, (1996).

23. Leithe, E. & Rivedal, E. Ubiquitination and Down-regulation of Gap Junction Protein Connexin-43 in Response to 12-O-Tetradecanoylphorbol 13-Acetate Treatment*. (2004) doi:10.1074/jbc.M402006200.
24. Settembre, C. *et al.* TFEB Links Autophagy to Lysosomal Biogenesis. *Science* (1979) **332**, 1429–1433 (2011).

Dear Henrique,

We have now received re-review reports from two referees. As you will see, you have addressed their concerns satisfactorily. Before I can finally accept the manuscript though, there are some remaining editorial points which need to be addressed. In this regard would you please:

- include acknowledgement of the following funders on our online submission system: European Regional Development Fund (ERDF) through the Operational Program for Competitiveness Factors (COMPETE) under the projects: HealthyAging2020 CENTRO-01-0145-FEDER-000012-N2323, CENTRO-01-0145- FEDER-032179, CENTRO-01-0145-FEDER-032414, PPBI-POCI-01-0145-FEDER-022122, UIDB/04539/2020, UIDP/04539/2020, project FCT-PTDC/MED-NEU/8030/2020, FCT-2022.09311, Synageing HR22-00854 from La Caixa Foundation, Horizon's grant 857524, and the Comissão de Coordenação e Desenvolvimento Regional do Centro - CCDRC through the Centro2020 Programme,
- add up to five keywords,
- change the reference format to an alphabetical list with 'et al.' used after 10 author names, remove DOIs from all references that are not preprints and datasets that have not been published yet,
- include a "Disclosure and competing interests statement",
- remove the AC/CrediT section from the text,
- remove the reference to unpublished data on p 15,
- remove figures from main text and provide as separate final production quality Figure files,
- use nomenclature Appendix Figure S1, etc. Appendix Table S1; the appendix file should begin with a table of contents with page numbers on the title page; all callouts in the manuscript text need to be correctly updated once the nomenclature is corrected in the Appendix file,
- consider including supplementary methods in the main manuscript file,
- rearrange the section order as follows: Title page - Abstract & Keywords - Introduction - Results - Discussion - Methods - Data Availability - Acknowledgments - Disclosure Statement & Competing Interests - References - Figure Legends - Tables with legends - Expanded View Figure Legends,
- rectify Movie 1 and 2 that are called out in the text but missing from submission,
- clarify a possible re-use of the image between Figure 2H (Ct / LAMP1) and Supp Fig 3G (Ct / LAMP1), stating in the figure legend if this control has been re-used,
- define the annotated p values *****/*/*** and provide the exact p-values for the same in the legend of figure 1c-d, g, i; 2a, d-g, i, l-m; 3c, g; 4a, d-f; 5b, d-e, g, i-j; 6b-c, f; 7b-c, e-f; as appropriate,
- indicate the statistical test used for data analysis in the legends of figures 7b-c, e-f,
- define terms of boxplots in the legend of figure 2g,
- define 'n' in the legend of figure 5b,
- describe the nature of entity 'n' in the legends of figures 1c-d, g, i; 2d-g; 3g; 4f; 5a; 7e-f,
- define error bars in the legend of figure 5b, and
- define scale bar for figures 2h, j.

We include a synopsis of the paper (see <http://emboj.embojournal.org/>). Please provide me with a general summary image; the size should be 550 pixels wide by 200-400 pixels high. You can also use something from the figures if that is easier. Please also supply a two-sentence summary statement and 3-5 bullet points that capture the key findings of the paper. I also need a one-sentence summary for our homepage.

EMBO Press is an editorially independent publishing platform for the development of EMBO scientific publications.

Best wishes,

William

William Teale, PhD
Editor
The EMBO Journal
w.teale@embojournal.org

- a point-by-point response to the referees' comments, with a detailed description of the changes made (as a word file).

- a word file of the manuscript text.

- individual production quality figure files (one file per figure)

- a complete author checklist, which you can download from our author guidelines

(<https://www.embopress.org/page/journal/14602075/authorguide>).

- Expanded View files (replacing Supplementary Information)

- a Reagents and Tools Table as part of the Methods section, which can be downloaded from our author guidelines

(<https://www.embopress.org/page/journal/14602075/authorguide#structuredmethods>)

We realize that it is difficult to revise to a specific deadline. In the interest of protecting the conceptual advance provided by the work, we recommend a revision within 3 months (27th Sep 2024). Please discuss the revision progress ahead of this time with the editor if you require more time to complete the revisions. Use the link below to submit your revision:

Referee #1:

My previous concerns have been addressed thoroughly with some new data and analysis.

Referee #2:

The Authors have satisfactorily addressed the concerns raised in my previous review.

All editorial and formatting issues were resolved by the authors.

Dear Henrique,

I am pleased to inform you that your manuscript has been accepted for publication in the EMBO Journal.

Congratulations! I am really excited to see this study in The EMBO Journal.

Best wishes,

William

William Teale, PhD
Editor
The EMBO Journal
w.teale@embojournal.org
